# TOWARDS ROBUST AND PARAMETER-EFFICIENT KNOWLEDGE UNLEARNING FOR LLMS

**Sungmin Cha**[1*] **Sungjun Cho**[2*] **Dasol Hwang**[3] **Moontae Lee**[3,4]
[1]New York University   [2]University of Wisconsin-Madison
[3]LG AI Research   [4]University of Illinois Chicago
sungmin.cha@nyu.edu, sungjuncho@cs.wisc.edu,
{dasol.hwang, moontae.lee}@lgresearch.ai

## ABSTRACT

Large Language Models (LLMs) have demonstrated strong reasoning and memorization capabilities via pretraining on massive textual corpora. However, this poses risk of privacy and copyright violations, highlighting the need for efficient machine unlearning methods that remove sensitive data without retraining from scratch. While Gradient Ascent (GA) is commonly used to unlearn by reducing the likelihood of generating unwanted content, it leads to unstable optimization and catastrophic forgetting of retrained knowledge. We find that combining GA with low-rank adaptation results in poor trade-offs between computational cost and generative performance. To address these challenges, we propose Low-rank Knowledge Unlearning (LoKU), a novel framework that enables robust and efficient unlearning for LLMs. First, we introduce Inverted Hinge Loss, which suppresses unwanted tokens while maintaining fluency by boosting the probability of the next most likely token. Second, we develop a data-adaptive initialization for LoRA adapters via low-rank approximation weighted with relative Fisher information, thereby focusing updates on parameters critical for removing targeted knowledge. Experiments on the Training Data Extraction Challenge dataset using GPT-Neo models as well as on the TOFU benchmark with Phi-1.5B and Llama2-7B models demonstrate that our approach effectively removes sensitive information while maintaining reasoning and generative capabilities with minimal impact. Our implementation can be found in https://github.com/csm9493/efficient-llm-unlearning.

## 1 INTRODUCTION

Large Language Models (LLMs) exhibit substantial performance gains in downstream tasks with increasing model size and amount of pretraining data (Zhao et al., 2023). This has prompted extensive research on collecting high-quality textual corpora for LLM pretraining and developing larger models to an unprecedented scale (Brown et al., 2020; Chowdhery et al., 2023; Smith et al., 2022; Rae et al., 2021; Dubey et al., 2024). However, this approach has introduced significant privacy concerns due to LLMs' tendency to memorize data indiscriminately (Carlini et al., 2021; 2023). For instance, Personally Identifiable Information (*e.g.*, names, phone numbers, and email addresses) can be easily extracted from LLMs (Carlini et al., 2021). Additionally, OpenAI is facing multiple copyright infringement lawsuits due to unpermitted use of licensed articles during LLM pretraining (Grynbaum & Mac, 2023). In response to such challenges as well as increasing interest in one's right to be forgotten (*e.g.*, the GDPR legislation) (Voigt & Von dem Bussche, 2017; Rosen, 2011; Villaronga et al., 2018), machine unlearning for LLMs has emerged a critical and rapidly growing research field (Yao et al., 2023; Si et al., 2023).

One method for LLM unlearning would be to filter out sensitive data from the corpus and retrain the model from scratch, an approach known as *exact* unlearning. With unprecedentedly large models and pretraining datasets, this process is highly resource-intensive and can easily become intractable under the possibility of multiple data deletion requests made in a sequential manner. This motivates *approximate* unlearning, where the goal is to remove knowledge of specific data instances without retraining the model from scratch (Figure 1). In this regard, several novel approaches have been

---

*equal contribution

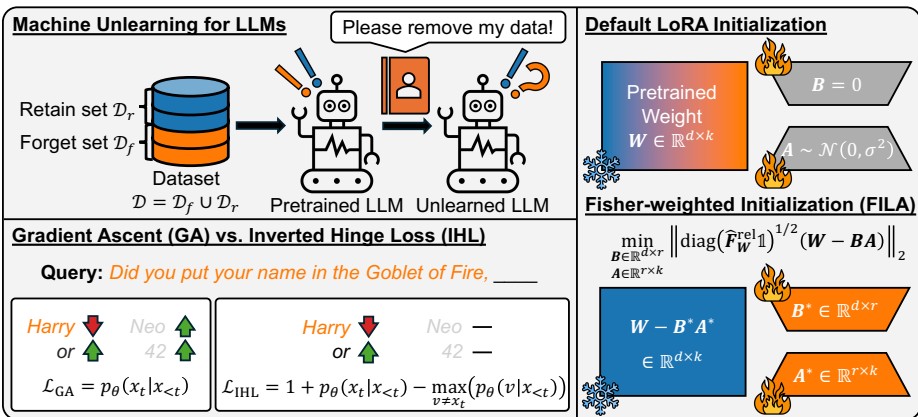

Figure 1: LLM unlearning aims to forget data points in $\mathcal{D}_f$ while maintaining knowledge of the retain set $\mathcal{D}_r$. Unlike GA, our IHL induces higher unlearning stability by reducing the likelihood of unwanted tokens in a controlled manner. To accelerate unlearning with IHL, FILA extracts and places parameters important in generating $\mathcal{D}_f$ to LoRA weights a priori via weighted low-rank approximation. IHL and FILA form a powerful synergy towards robust and efficient LLM unlearning.

proposed for approximate unlearning: Jang et al. (2023) introduced a simple method that finetunes LLMs using Gradient Ascent (GA) on data requested for deletion and also proposed $n$-gram-based metrics to evaluate its effectiveness. Wang et al. (2023) and Liu et al. (2024) proposed knowledge distillation-based methods that selectively transfer knowledge to a secondary model for unlearning. However, both approaches face significant challenges: GA suffers from unstable optimization due to unbounded nature of its objective loss, while distillation-based methods incur substantial computational costs from relying on a secondary model. Above all, these approaches share a critical drawback: the high computational cost of full fine-tuning all parameters within the LLMs.

Meanwhile, Low-Rank Adaptation (LoRA) has emerged as one of the most prominent techniques for parameter-efficient fine-tuning on downstream tasks (Hu et al., 2022). The core idea of LoRA is to freeze all pretrained weights and instead train low-rank decomposition matrices to model the weight changes in each linear layer, effectively reducing the number of trainable parameters and thus its memory cost. Beyond its efficiency, the low-rank structure in LoRA also serves as a strong regularizer (Biderman et al., 2024), which we hypothesize aids LLM unlearning by stabilizing optimization and mitigating catastrophic forgetting of retained knowledge. However, the empirical effects of LoRA in the context of LLM unlearning remain largely unexplored.

In this paper, we present the first in-depth study of LLM unlearning under the low-rank adaptation paradigm and introduce Low-rank Knowledge Unlearning (LoKU), which consists of two novel techniques for robust and parameter-efficient knowledge unlearning. First, we analyze the derivatives of GA to highlight its drawbacks such as (1) unnecessary forgetting induced by indiscriminately increasing the probability of all tokens and (2) unstable optimization due to the unboundedness of maximizing the next-token prediction loss. To tackle these challenges, we introduce the Inverted Hinge Loss (IHL), which replaces each unwanted token with the next most-probable one, and demonstrate that IHL facilitates stable unlearning by resolving the limitations of GA. Second, we find that the low-rank regularization in LoRA is too strong when unlearning with IHL, leading to suboptimal cost vs. post-unlearning performance trade-offs. In response, we propose Fisher-Initialization of Low-rank Adapters (FILA), which data-adaptively assigns parameters responsible for generating unwanted information to adapters prior to tuning by decomposing the pretrained parameters weighted by the relative Fisher-information matrix (Figure 1). Experiments on the Training Data Extraction Challenge dataset (GPT-Neo) and the TOFU benchmark (Phi-1.5B, Llama2-7B) demonstrate that IHL combined with FILA outperforms existing baselines in both efficiency and post-unlearning performance. In summary, our main contributions are as follows:

- We analyze the shortcomings of GA—unbounded optimization and unnecessary forgetting—through its derivative and propose IHL to address these issues.

- We introduce FILA, a method to accelerate unlearning by data-adaptively assigning parameters responsible for unwanted information to low-rank adapters before tuning.

- We demonstrate that our proposed LoKU using both IHL and FILA outperforms previous baselines in terms of both efficiency and post-unlearning performance (Figure 2).

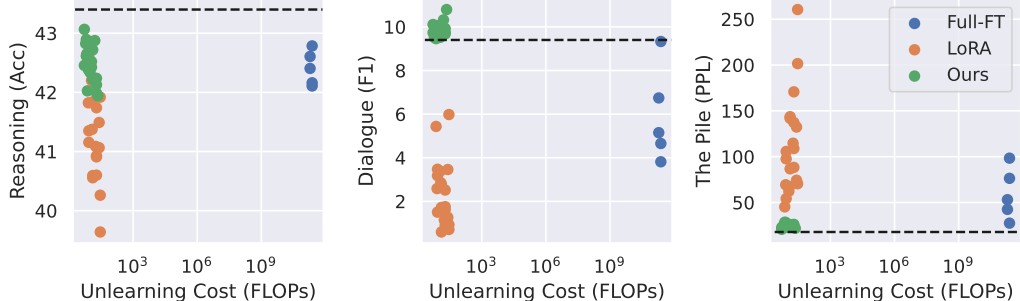

Figure 2: Compute cost for successful unlearning vs. post-unlearning downstream performances. We unlearn 32 randomly sampled sequences in the Training Data Extraction Challenge from GPT-Neo-125M. Each point represents a different forget set and LoRA rank (if used). **Left:** Accuracy averaged across 9 classification tasks (higher is better). **Middle:** F1 score averaged across 4 dialogue generation tasks (higher is better). **Right:** Perplexity on the validation set of the Pile dataset (lower is better). Dashed lines indicate the performances of the model prior to unlearning. Unlearning via gradient differences (GD) with vanilla LoRA leads to significant loss in performance compared to full-parameter GD unlearning due to lack of plasticity. However, our proposed LoKU using both the Inverted Hinge Loss and Fisher-weighted LoRA initialization performs competitively to unlearning via full-finetuning in all three aspects while enjoying the parameter-efficiency of LoRA.

## 2 RELATED WORK

**Machine Unlearning.** The primary objective of machine unlearning is to adapt a pretrained model to discard information acquired from a specific subset of data previously used during pretraining, with active research focused on image classification (Cao & Yang, 2015; Golatkar et al., 2020; Tarun et al., 2023; Mehta et al., 2022; Chundawat et al., 2023; Cha et al., 2024). Recently, its significance has grown notably with Large Language Models (LLMs) due to crucial need for managing unintended memorization of pretraining data intrinsic to LLMs (Si et al., 2023; Yao et al., 2024b). Several machine unlearning algorithms for LLMs focus on parameter optimization (Si et al., 2023). For example, Wang et al. (2023) introduced Knowledge Gap Alignment, using knowledge distillation between models trained on different datasets. Chen & Yang (2023) proposed an unlearning layer to selectively remove specific knowledge while preserving other parameters, while Liu et al. (2024) developed a two-stage framework to capture and negate harmful knowledge. However, these approaches are limited by the need to retain large datasets (Wang et al., 2023; Chen & Yang, 2023) or rely on secondary models for distillation (Wang et al., 2023; Liu et al., 2024). Model-editing methods, such as task arithmetic for suppressing harmful content (Ilharco et al., 2023; Wu et al., 2023), avoid substantial costs but show limited effectiveness in unlearning. In contrast, Jang et al. (2023) used Gradient Ascent (GA) for LLM unlearning by maximizing the next-token prediction loss on the forget data, effectively unlearning while preserving model performance. Following this, GA has become a standard baseline, with Yao et al. (2024a) improving its robustness by combining GA with gradient descent on in-distribution data.

**Parameter-Efficient Fine-Tuning.** Fine-tuning LLMs for specific tasks is computationally expensive due to their large parameter sizes. To address this, Parameter-Efficient Fine-Tuning (PEFT) methods adapt only a small subset of parameters while keeping the pretrained ones frozen (Liu et al., 2022b; Qiu et al., 2023; Liu et al., 2023). Inspired by the small intrinsic rank of LLMs (Li et al., 2018; Aghajanyan et al., 2021), LoRA and its derivatives add low-rank adapters to the model's linear layers (Hu et al., 2022; Zhang et al., 2023; Yeh et al., 2023; Kopiczko et al., 2024; yang Liu et al., 2024). These adapters can be merged with pretrained parameters after fine-tuning, maintaining the original inference cost. While most methods use random initialization for LoRA adapters, PiSSA (Meng et al., 2024) suggests initializing them using the principal singular vectors and values of the linear weights.

Although various methods in machine unlearning and parameter-efficient fine-tuning for LLMs have been discussed, this paper focuses on analyzing the inherent issues of GA, introducing a novel unlearning loss function to overcome these issues, and exploring parameter-efficient unlearning methods that do not require full fine-tuning. These areas have not been adequately addressed in previous studies, highlighting the contributions of our work.

# 3 PROPOSED METHOD: LOW-RANK KNOWLEDGE UNLEARNING (LoKU)

## 3.1 PRELIMINARIES

**Problem and Notation.** Given a sequence of $T$ tokens $\boldsymbol{x} = (x_1, x_2, \ldots, x_T)$, a language model (LM) models the likelihood of the sequence via next-token prediction: $p_\theta(\boldsymbol{x}) = \prod_{t=1}^{T} p_\theta(x_t|x_{<t})$. After pretraining, we assume that an end-user has requested to delete a subset of the training set $\mathcal{D}_f \subset \mathcal{D}$, which we refer to as the *forget set*. The *retain set* $\mathcal{D}_r$ refers to an auxiliary dataset that contains other relevant knowledge that must be retained after unlearning (*e.g.*, Wikitext; Merity et al. 2017).

**Gradient Ascent.** Ideally, the LM must assign low probability to sequences in $\mathcal{D}_f$, leading to a simple yet effective baseline of Gradient Ascent (GA; Jang et al. 2023). GA unlearns a sequence of tokens $\boldsymbol{x} = (x_1, \ldots, x_T)$ by maximizing the next-token prediction loss

$$\mathcal{L}_{\text{GA}}(\boldsymbol{x}) = -\sum_{t=1}^{T} \log(p_\theta(x_t|x_{<t})) \tag{1}$$

In practice, the log-likelihood is computed using cross-entropy loss and thus GA essentially minimizes the Negative Cross-Entropy (NCE) loss. Therefore, GA maximizing the next-token prediction loss involves unbounded optimization, leading to an ill-posed process with unstable tuning. While Gradient Difference (GD) aims to alleviate this instability by minimizing the next-token prediction loss for $\mathcal{D}_r$ alongside NCE on $\mathcal{D}_f$ as regularization, we find that the approach falls short of a fundamental solution, showing performance degradation as unlearning updates are made.

**Low-Rank Adaptation.** Based on the assumption that parameter changes due to LLM adaptation exhibits an intrinsic low-rank (Aghajanyan et al., 2021), LoRA models the change in parameters $\Delta \boldsymbol{W} \in \mathbb{R}^{d \times k}$ of each linear weight $\boldsymbol{W} \in \mathbb{R}^{d \times k}$ via a product of two low-rank matrices $\boldsymbol{A} \in \mathbb{R}^{r \times k}$ and $\boldsymbol{B} \in \mathbb{R}^{d \times r}$ where $r \ll \min(d, k)$ is the rank of the LoRA adapter. In other words, the output of the adapted linear layer given an input $\boldsymbol{x}$ becomes

$$(\boldsymbol{W} + \Delta \boldsymbol{W})\boldsymbol{x} = \boldsymbol{W}\boldsymbol{x} + \boldsymbol{B}\boldsymbol{A}\boldsymbol{x}$$

During fine-tuning, the original weight $\boldsymbol{W}$ is kept frozen and only the low-rank factors $\boldsymbol{A}$ and $\boldsymbol{B}$ are updated via gradient descent. To ensure that the initial attachment of LoRA adapters does not alter the output of the LLM, LoRA defaults to initializing $\boldsymbol{A}$ with a Kaiming-uniform distribution (He et al., 2015) and $\boldsymbol{B}$ as the zero matrix. After finetuning, LoRA adapters can simply be merged with the original weights $\boldsymbol{W}' = \boldsymbol{W} + \boldsymbol{B}\boldsymbol{A}$, thereby avoiding any additional latency during inference.

## 3.2 PRELIMINARY RESULTS

Despite its wide use in domain adaptation and instruction tuning, LoRA is not yet explored under the task of LLM unlearning to the best of our knowledge. Therefore, we first share empirical results from low-rank adapting LLMs using GD as our objective to motivate our approach. Figure 2 shows the results. Notably, vanilla LoRA suffers from lack of plasticity and ends up failing to sufficiently unlearn $\mathcal{D}_f$ within 20 epochs. When running more unlearning epochs or increasing the learning rate for sufficient unlearning, the model loses its previously acquired reasoning and generative capabilities, as shown in the significant decrease in Reasoning and Dialogue performances. In the remainder of this section, we present two techniques towards making LLM unlearning viable while enjoying the efficiency of LoRA.

## 3.3 INVERTED HINGE LOSS: A NOVEL LOSS FUNCTION FOR LLM UNLEARNING

**Motivation.** We analyze the inherent issues of GA from the perspective of its derivative. The output layer of a language model is a softmax layer that outputs probabilities over the vocabulary. Let $y_t$ be the logits (pre-softmax activations) produced by the LLM model for the $t$-th token, and let $V$ be the vocabulary size. The probability $p_\theta(x_t|x_{<t})$ is given by the softmax function: $p_\theta(x_t|x_{<t}) = \exp(y_t^{(x_t)})/\sum_{v=1}^{V} \exp(y_t^{(v)})$ where $y_t^{(x_t)}$is the logit corresponding to the true token $x_t$ and $y_t^{(v)}$ is the logit corresponding to the $v$-th token in the vocabulary. When we use $\mathcal{L}_{\text{GA}}$ for unlearning for LLMs, the gradient of the log-probability with respect to the logits is

$$\frac{\partial \log(p_\theta(x_t|x_{<t}))}{\partial y_t^{(v)}} = \begin{cases} 1 - p_\theta(x_t|x_{<t}) & \text{if } v = x_t \\ -p_\theta(v|x_{<t}) & \text{if } v \neq x_t \end{cases}$$

From this derivative of GA, we can interpret its unlearning mechanism: given the prefix $x_{<t}$, GA reduces the prediction score of the true token $x_t$ in proportion to $1 - p_\theta(x_t|x_{<t})$ while increasing the scores of other tokens (*i.e.*, $v \neq x_t$) by $p_\theta(v|x_{<t})$. This process effectively shifts the model's prediction for $x_{<t}$ away from the true token $x_t$, thereby achieving unlearning. However, we can confirm that GA suffers form the following problems during unlearning: (1) **Gradient spread**, where reducing the score of $x_t$ while increasing the scores of all other tokens leads to inefficient unlearning in large vocabularies by predominantly boosting other tokens; (2) **Unbounded loss**, where minimizing $\log(p_\theta(x_t|x_{<t}))$ through maximizing cross-entropy loss introduces a risk of divergence due to the unbounded nature of entropy; and (3) **Degradation of generative performance**, where GA applies uniform gradient updates (*i.e.* increasing the scores of other tokens) to all sequences in the forget set $\mathcal{D}_f$, despite each sequence requiring a unique number of updates for unlearning. This redundancy can cause degrade the model's generative capabilities, resulting in catastrophic forgetting.

**Inverted Hinge Loss.** To cope with aforementioned limitations of GA, we aim to design a new loss function that achieves effective unlearning by decreasing the prediction score of the true token, while focusing gradient updates on only a minimal number of viable replacements for the ground-truth token. Inspired by the Hinge Loss (Cortes & Vapnik, 1995), we devise Inverted Hinge Loss (IHL) as

$$\mathcal{L}_{\text{IHL}}(\boldsymbol{x}) = 1 + p_\theta(x_t|x_{<t}) - \max_{v \neq x_t}(p_\theta(v|x_{<t}))$$

As the probability $p_\theta(x_t|x_{<t})$ is given by the softmax function, the derivative of $\mathcal{L}_{\text{IHL}}(\boldsymbol{x})$ with respect to $y_t^{(v)}$ is:

$$\frac{\partial \mathcal{L}_{\text{IHL}}(\boldsymbol{x})}{\partial y_t^{(v)}} = \begin{cases} p_\theta(x_t|x_{<t})(p_\theta(v^\star|x_{<t}) - p_\theta(x_t|x_{<t}) + 1) & \text{if } v = x_t \\ p_\theta(v^\star|x_{<t})(p_\theta(v^\star|x_{<t}) - p_\theta(x_t|x_{<t}) - 1) & \text{if } v = v^\star \\ p_\theta(v|x_{<t})(p_\theta(v^\star|x_{<t}) - p_\theta(x_t|x_{<t})) & \text{if } v \neq x_t \text{ and } v \neq v^\star, \end{cases}$$

where $v^\star = \arg\max_{v \neq x_t} p_\theta(v|x_{<t})$. The detailed derivation can be found in Appendix A.

The above derivative clearly illustrates how the IHL addresses the shortcomings of GA in knowledge unlearning for LLMs. First, IHL mitigates **gradient spread** by ensuring that the gradients primarily focus on the true token $x_t$ and its competitive token $v^\star$, without excessively boosting irrelevant tokens in large vocabularies. For example, in the case where unlearning has not yet been achieved (*i.e.*, when $p_\theta(x_t|x_{<t}) > p_\theta(v^\star|x_{<t})$), the absolute value of the gradient for the true token $x_t$ is equal to or greater than that of $v^\star$ (with opposite sign) and exceeds that of other tokens (*i.e.*, $v \neq x_t$ and $v \neq v^\star$). This ensures efficient and targeted unlearning while avoiding the unnecessary spread of gradients across irrelevant tokens. Second, IHL resolves the issue of **unbounded loss** by defining a bounded loss function, ensuring that the prediction scores for tokens other than $x_t$ and $v^\star$ decrease once unlearning is complete (*i.e.*, when $p_\theta(x_t|x_{<t})$ becomes less than $p_\theta(v^\star|x_{<t})$). This bounded loss prevents instability and divergence during unlearning. Lastly, IHL prevents the **degradation of generative performance** by adapting gradient updates based on the prediction scores of $x_t$ and $v^\star$, alleviating redundant updates for other tokens (*i.e.*, $v \neq x_t$ and $v \neq v^\star$). Compared to GA, this approach reduces redundant updates for other tokens in each sequence of the forget set $\mathcal{D}_f$, thereby preserving the model's generative capabilities while achieving effective and stable unlearning.

### 3.4 FILA: A NOVEL LoRA INITIALIZATION FOR LLM UNLEARNING

**Motivation.** While IHL effectively stabilizes the unlearning process by reducing the likelihood of unwanted tokens in a controlled manner, we empirically find that combining IHL with LoRA naïvely leads to requiring large number of unlearning iterations to fully forget samples in $\mathcal{D}_f$. Naturally, an increasing number of updates with $\mathcal{D}_f$ comes with the risk of overfitting, which for unlearning, may result in significant loss of knowledge on $\mathcal{D}_r$. We hypothesize that this is due to the random initialization of LoRA weights together with its low-rank structure imposing too strong a regularization to handle precise gradients needed for proper minimization of IHL (Biderman et al., 2024). Drawing inspiration from PiSSA (Meng et al., 2024), we therefore aim to accelerate optimization of IHL under LoRA by extracting weights relatively more important to $\mathcal{D}_f$ than to $\mathcal{D}_r$ from each pretrained weight, then using them to initialize LoRA weights prior to unlearning. We conjecture that this approach reinforces the model's plasticity to forget $\mathcal{D}_f$ as well as its stability on keeping knowledge on $\mathcal{D}_r$. The remainder of this section presents how parameter importances are measured via Fisher information, and how low-rank adapter weights are initialized based on the measured importances.

**Parameter Importances via Fisher Information.** The Fisher information matrix $\boldsymbol{F}_\theta(\mathcal{D})$ is defined as the variance of the partial derivative of the log-likelihood of data $\mathcal{D}$ with respect to the model parmeter $\theta$ (left of Equation 2). Intuitively, the matrix can be considered a measurement on how much the model output changes following a small change on its parameter weight. However, as marginalizing across the space of $\mathcal{D}$ is intractable, many works in continual learning (Kirkpatrick et al., 2017) and model compression (Hsu et al., 2022) literature have thus used the empirical Fisher information $\hat{\boldsymbol{F}}_\theta(\mathcal{D})$ instead. In the context of LLMs, this can be computed as:

$$\boldsymbol{F}_\theta(\mathcal{D}) = \mathbb{E}_\mathcal{D}\left[\left(\frac{\partial}{\partial\theta}\log p_\theta(\mathcal{D}|\theta)\right)^2\right] \approx \frac{1}{|\mathcal{D}|}\sum_{\boldsymbol{x}\in\mathcal{D}}\left(\frac{\partial}{\partial\theta}\mathcal{L}_{\text{LM}}(\boldsymbol{x};\theta)\right)^2 =: \hat{\boldsymbol{F}}_\theta(\mathcal{D}), \qquad (2)$$

where $\mathcal{L}_{\text{LM}}$ denotes the next-token prediction loss used to pretrain LMs, $\mathcal{L}_{\text{LM}}(\boldsymbol{x};\theta) = \sum_{t=1}^{T}\log(p_\theta(x_t|x_{<t}))$. Within our LLM unlearning setup, a high empirical Fisher information measured with $\mathcal{D}_f$ indicates that $\mathcal{L}_{\text{LM}}$ on $\mathcal{D}_f$ leads to large absolute gradients on the parameter under concern, and we consider such parameters to be *important* in generating sequences in $\mathcal{D}_f$.

Let $\hat{\boldsymbol{F}}_{\boldsymbol{W}}^f := \hat{\boldsymbol{F}}_{\boldsymbol{W}}(\mathcal{D}_f)$ denote the empirical Fisher information matrix of the target parameter $\boldsymbol{W}$ measured using the forget set $\mathcal{D}_f$ (resp. $\hat{\boldsymbol{F}}_{\boldsymbol{W}}^r$ using the retain set $\mathcal{D}_r$). Then, we use the relative Fisher information $\hat{\boldsymbol{F}}_{\boldsymbol{W}}^{\text{rel}} := \hat{\boldsymbol{F}}_{\boldsymbol{W}}^f/\hat{\boldsymbol{F}}_{\boldsymbol{W}}^r \in \mathbb{R}^{d\times k}$ as an importance metric to identify parameters that are important exclusively for $\mathcal{D}_f$ and not for $\mathcal{D}_r$. While generating $\mathcal{D}_f$ involves extracting memorized information on $\mathcal{D}_f$ as well as composing linguistically fluent outputs, we only wish to adjust parameters responsible for the former and thus use $\hat{\boldsymbol{F}}_{\boldsymbol{W}}^{\text{rel}}$ rather than $\hat{\boldsymbol{F}}_{\boldsymbol{W}}^f$.

**Fisher-weighted Initialization of Low-rank Adapters.** Given the relative importance $\hat{\boldsymbol{F}}_{\boldsymbol{W}}^{\text{rel}}$ for each target weight $\boldsymbol{W}$, we propose to initialize the corresponding LoRA adapter weights with the solution to the following Weighted Low-Rank Approximation (WLRA) problem:

$$\min_{\boldsymbol{A}\in\mathbb{R}^{r\times k},\boldsymbol{B}\in\mathbb{R}^{d\times r}}\sum_{i,j}\left([\hat{\boldsymbol{F}}_{\boldsymbol{W}}^{\text{rel}}]_{i,j}(\boldsymbol{W}-\boldsymbol{BA})_{i,j}\right)^2.$$

Note that when all weights $[\hat{\boldsymbol{F}}_{\boldsymbol{W}}^{\text{rel}}]_{i,j}$ equal one, WLRA reduces to standard low-rank matrix approximation, for which the solution can easily be computed via rank-$r$ SVD. For general weights, however, this minimization problem does not have a closed-form solution and requires iterative optimization (Srebro & Jaakkola, 2003). While we may resort to iterative methods to initialize LoRA weights in a fine-grained manner, this would undermine the efficiency gains from deploying low-rank adapters. Therefore, we assume that parameters in each row of $\boldsymbol{W}$ share the same importance equal to the square-root of the row-wise sum of $\hat{\boldsymbol{F}}_{\boldsymbol{W}}^{\text{rel}}$, and simplify the problem to

$$\min_{\boldsymbol{A}\in\mathbb{R}^{r\times k},\boldsymbol{B}\in\mathbb{R}^{d\times r}}\left\|\texttt{diag}\left((\hat{\boldsymbol{F}}_{\boldsymbol{W}}^{\text{rel}}\mathbb{1})^{\frac{1}{2}}\right)(\boldsymbol{W}-\boldsymbol{BA})\right\|_2$$

with $\mathbb{1}\in\mathbb{R}^k$ and $\texttt{diag}(\cdot)$ indicating the all-one vector and the vector diagonalization function, respectively. Unlike general WLRA, this row-wise WLRA problem has a closed-form solution, which can be obtained by applying rank-$r$ SVD to decompose $\texttt{diag}((\hat{\boldsymbol{F}}_{\boldsymbol{W}}^{\text{rel}}\mathbb{1})^{\frac{1}{2}})\boldsymbol{W} = \boldsymbol{USV}^T$ and computing $\boldsymbol{B}^* = (\hat{\boldsymbol{F}}_{\boldsymbol{W}}^{\text{rel}}\mathbb{1})^{-\frac{1}{2}}\boldsymbol{US}^{\frac{1}{2}}$ and $\boldsymbol{A}^* = \boldsymbol{S}^{\frac{1}{2}}\boldsymbol{V}^T$.

Given this solution, we use $\boldsymbol{B}^*$ and $\boldsymbol{A}^*$ as initial LoRA weights. To ensure that the model behavior remains the same after LoRA initialization, the base layers are also updated with $\boldsymbol{W}^* = \boldsymbol{W} - \boldsymbol{B}^*\boldsymbol{A}^*$. Intuitively, our Fisher-weighted Initialization of Low-rank Adapters (FILA) extracts parameters that are important for generating $\mathcal{D}_f$ but not for generating $\mathcal{D}_r$, such that LoRA tuning can be focused on erasing knowledge relevant to $\mathcal{D}_f$ while keeping information regarding $\mathcal{D}_r$.

## 3.5 FINAL LOSS FUNCTION OF LoKU

In summary, we perform unlearning on the model $\boldsymbol{\Theta} = \boldsymbol{\theta} \cup \boldsymbol{\theta}_{\text{FILA}}$ consisted of original pretrained weights $\boldsymbol{\theta}$ and FILA-initialized low-rank adapter weights for each linear layer $\boldsymbol{\theta}_{\text{FILA}} = \{\boldsymbol{A}_\ell^*, \boldsymbol{B}_\ell^*\}_{\ell=1}^L$, where $L$ represents the number of layers tuned via LoRA. Additionally, we incorporate GD, which utilizes the auxiliary retain set $\mathcal{D}_r$. The final loss function of LoKU is defined as follows:

$$\underset{\theta_{\text{FILA}}}{\text{minimize}}\sum_{\boldsymbol{x}_r\in\mathcal{D}_f,\boldsymbol{x}_f\in\mathcal{D}_r}\mathcal{L}_{\text{IHL}}(\boldsymbol{x}_f) + \mathcal{L}_{\text{LM}}(\boldsymbol{x}_r) \qquad (3)$$

In practice, training (unlearning) for the LLM model is conducted by minimizing Equation 3 through stochastic gradient descent.

Table 1: Evaluation results on reasoning and generative capabilities before and after unlearning samples from the TDEC dataset. For each model, we test three unlearning objectives (*i.e.* GA, GD, IHL) with full-parameter tuning, and four methods using LoRA with rank 16. IHL denotes replacing the GA loss term in GD with our IHL, and +FILA denotes initializing LoRA weights with FILA. Changes in performance after unlearning are in parentheses, with best results under LoRA-based unlearning in **bold**.

| Model | Method | Params. (%) | Epochs | $\text{EL}_{10}$ (%)↓ | MA (%)↓ | Reasoning (Acc)↑ | Dialogue (F1)↑ | Pile (PPL)↓ |
|---|---|---|---|---|---|---|---|---|
| GPT-Neo 125M | Before | - | - | 30.9 | 77.4 | 43.4 | 9.4 | 17.8 |
| | GA | 100.0 | 17.2 | 1.0 | 27.4 | 39.9 (-3.5) | 2.6 (-6.8) | 577.8 (+560.0) |
| | GD | | 4.6 | 0.7 | 24.9 | 42.4 (-1.0) | 5.9 (-3.5) | 54.2 (+36.4) |
| | IHL | | 17.2 | 0.7 | 29.2 | 42.3 (-1.1) | 10.3 (+0.9) | 18.1 (+0.3) |
| | GD | 1.6 | 8.6 | 0.3 | 20.6 | 40.8 (-2.6) | 2.5 (-6.9) | 129.4 (+111.6) |
| | IHL | | 11.4 | 0.4 | 22.7 | 41.9 (-1.5) | 6.0 (-3.4) | 32.9 (+15.1) |
| | GD+FILA | | 7.4 | 1.2 | 27.4 | 42.0 (-1.4) | 6.5 (-2.9) | 89.5 (+71.7) |
| | LoKU (IHL+FILA) | | 6.0 | 0.3 | 23.9 | **42.2 (-1.2)** | **10.1 (+0.7)** | **24.0 (+6.2)** |
| GPT-Neo 1.3B | Before | - | - | 67.6 | 92.2 | 49.8 | 11.5 | 11.5 |
| | GA | 100.0 | 13.8 | 1.9 | 30.4 | 49.7 (-0.1) | 8.5 (-3.0) | 15.8 (+4.3) |
| | GD | | 12.8 | 2.2 | 30.9 | 48.4 (-1.4) | 12.7 (+1.2) | 10.8 (-0.7) |
| | IHL | | 7.6 | 0.7 | 30.4 | 48.4 (-1.4) | 12.5 (+1.0) | 11.0 (-0.5) |
| | GD | 0.8 | 19.3 | 1.7 | 31.4 | 45.0 (-4.8) | 9.7 (-1.8) | 31.8 (+20.3) |
| | IHL | | 20.0 | 1.7 | 44.6 | 47.1 (-2.7) | 10.2 (-1.3) | 14.9 (+3.4) |
| | GD+FILA | | 7.8 | 1.9 | 23.2 | 44.2 (-5.6) | 5.5 (-6.0) | 54.5 (+43.0) |
| | LoKU (IHL+FILA) | | 13.0 | 0.5 | 29.6 | **48.3 (-1.5)** | **12.1 (+0.6)** | **14.7 (+3.2)** |
| GPT-Neo 2.7B | Before | - | - | 70.4 | 93.4 | 52.3 | 11.5 | 10.4 |
| | GA | 100.0 | 10.8 | 1.6 | 31.0 | 51.9 (-0.4) | 11.1 (-0.4) | 17.9 (+7.5) |
| | GD | | 8.0 | 0.7 | 28.3 | 51.8 (-0.5) | 12.7 (+1.2) | 17.9 (+7.5) |
| | IHL | | 6.6 | 0.5 | 29.3 | 51.8 (-0.5) | 12.9 (+1.4) | 10.7 (+0.3) |
| | GD | 0.7 | 14.0 | 0.1 | 20.4 | 45.9 (-6.4) | 6.7 (-4.8) | 61.1 (+50.7) |
| | IHL | | 17.8 | 0.0 | 26.7 | **49.6 (-2.7)** | 8.5 (-2.6) | 22.2 (+11.8) |
| | GD+FILA | | 6.8 | 1.6 | 28.9 | 44.8 (-7.5) | 9.3 (-2.2) | 68.7 (+58.3) |
| | LoKU (IHL+FILA) | | 10.3 | 0.1 | 28.5 | **49.6 (-2.7)** | **10.7 (-0.8)** | **16.0 (+5.6)** |

## 4 EXPERIMENTS

In this section, we first perform experiments unlearning samples from the Training Data Extraction Challenge (TDEC; §4.1), followed by ablation and analytical results (§4.2). We also conduct experiments on the Task of Fictitious Unlearning (TOFU; §4.3), a benchmark that well-mimics a real-world scenario for LLM unlearning evaluation. For brevity, we present results from additional experiments such as continual unlearning in Appendix D.

### 4.1 TRAINING DATA EXTRACTION CHALLENGE

**Experimental Setup.** The Training Data Extraction Challenge (TDEC) dataset (Carlini et al., 2021) consists of 20k examples from the Pile dataset (Gao et al., 2020) found to be easily extractable from a pretrained LLM. For each experiment, we randomly sample 32 sequences with 200 tokens to consist the forget set $\mathcal{D}_f$. For the retain set $\mathcal{D}_r$, we use the subset of WikiText (Merity et al., 2017) as it contains factual world knowledge that we wish to maintain after unlearning. We consider GPT-Neo 125M, 1.3B, and 2.7B pretrained on the Pile dataset as our base models, and unlearn $\mathcal{D}_f$ using five different forget sets. For this experiment, we use a fixed learning rate of 2e-4 and use LoRA adapters with rank $r = \{4, 8, 16, 32\}$. For reasons we illustrate later in §4.2, we choose to apply LoRA on query and value layers in the attention module and two linear layers within feed-forward layers.

Following previous work (Jang et al., 2023), we measure the unlearning efficacy with two metrics. The **$n$-gram Extraction Likelihood ($\text{EL}_n$)** measures the $n$-gram overlap between the ground truth sequence in $\mathcal{D}_f$ and the output generated by the model. The **Memorization Accuracy (MA)** measures the token-wise accuracy of the LLM on $\mathcal{D}_f$. More details on these metrics are shared in Appendix B. After each unlearning epoch, we measure $\text{EL}_{10}$ and MA of the model, and we consider the model has successfully unlearned $\mathcal{D}_f$ if both values measured on $\mathcal{D}_f$ become smaller than those measured from a held-out validation set that the model has never seen before within 20 unlearning epochs. Once unlearning is finished, we evaluate the unlearned model on various downstream benchmarks to measure how well the LLM maintains its previously acquired reasoning and generative capabilities. To assess its reasoning capabilities, we average accuracies across 9 different classification datasets. To measure generative performance, we also average the F1 scores over four dialogue generation datasets. Lastly, we measure the perplexity on the validation subset of the Pile (Gao et al., 2020). A comprehensive list of evaluation datasets can be found in Appendix C.

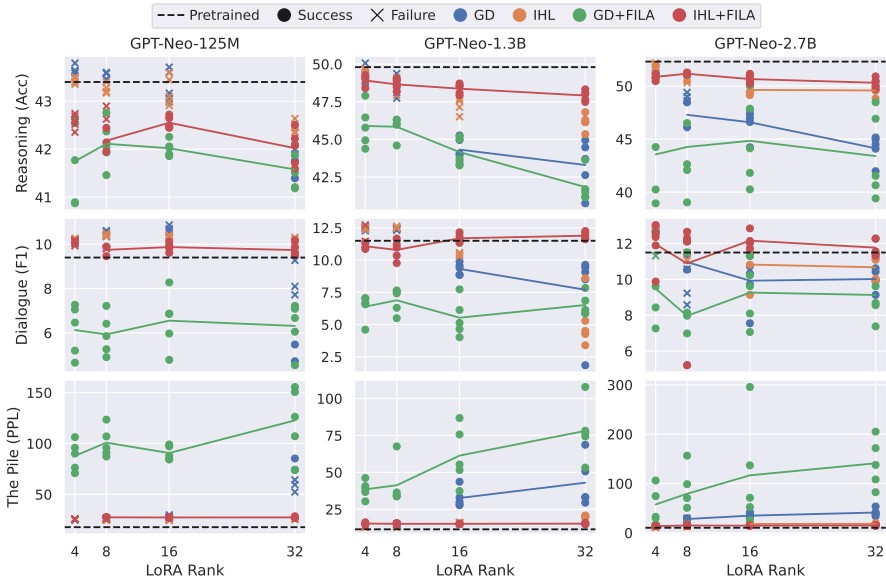

Figure 3: Results from unlearning examples in the TDEC dataset on the GPT-Neo LLM family. Each row represents the performance averaged across datasets within each set of LLM capability tests: Reasoning (higher is better), Dialogue (higher is better), and Perplexity (lower is better). The circles and crosses represent successful and unsuccessful attempts, respectively, of unlearning a particular forget set $\mathcal{D}_f$. Solid lines indicate the performance of different methods averaged *only across successful unlearning trials*. The dashed lines indicate the base model performance prior to unlearning. GD leads to significant loss in performance and also fails to unlearn in some cases even with large LoRA ranks. Replacing the NCE loss in GD with IHL boosts retention of reasoning and generation capabilities, but still fails to unlearn in multiple cases. Running GD with FILA notably increases the rate of unlearning success, but at significant cost in overall performance. Our LoKU using both IHL and FILA best minimizes post-unlearning performance degradation in all aspects.

We consider two LLM unlearning baselines, Gradient Ascent (GA) (Jang et al., 2023) and Gradient Difference (GD) (Liu et al., 2022a; Maini et al., 2024), both of which only require the original language model and datasets $\mathcal{D}_f$ and $\mathcal{D}_r$ representing knowledge we wish to unlearn and retain, respectively. We exclude methods that require another auxiliary model (Wang et al., 2023; Liu et al., 2024) or the entire training data (Wang et al., 2023; Chen & Yang, 2023) from our baselines.

**Results.** Table 1 shows evaluation results using a fixed LoRA rank of 16 and Figure 3 shows analogous results using a different LoRA ranks. Our key findings are as follows. First, using GD not only meets the forgetting criteria in fewer epochs across all model sizes but it also preserves previously acquired knowledge (*i.e.*, performance on Reasoning, Dialogue, and Pile) better than GA. Mainly for 125M and 1.3B models, GA causes a significant decline in generative performance, whereas GD partially mitigates this decline and improves both Dialogue F1 scores and the Pile perplexity. Second, we find that simply applying GD with LoRA fails to unlearn effectively across all model sizes. While enjoying great parameter-efficiency by tuning only about 0.7% to 1.6% of the total parameters when using LoRA, its application to unlearning with GD results in large loss in overall language capability, especially on generative tasks. Third, replacing GA in GD with IHL leads to performance gains in both full-parameter and LoRA-based unlearning, but requires larger number of epochs for successful unlearning than GD especially when confined to low-rank weight changes. However, this is resolved when IHL is used together with FILA, which significantly reduces the number of required epochs. Note that using FILA with GD also reduces the number of epochs required, but worsens downstream performance in many cases, as FILA essentially accelerates the divergent behavior of GA in GD. In essence, whether the ability of FILA to accelerate tuning also translates to benefits in downstream performance depends on the loss function being optimized. On the other hand, when paired with our bounded IHL (*i.e.*, LoKU), FILA leads to better retention of LLM performance, showcasing the strong synergy of our approach that enjoys not only the stability of IHL but also the speedup from FILA.

## 4.2 ANALYSIS

**What modules do we need to adapt?** Figure 4 presents experiments where low-rank adapters are attached to various target parameter groups, including those for Query (Q), Value (V), Key (K), Output (O) in the attention module, and the Feed-Forward Network (FFN). While the original LoRA

Figure 4: Results from unlearning examples from TDEC dataset using LoRA with rank 32 to adapt sets of layers on GPT-Neo-125M. The marker shapes and colors are used similarly as in Figure 3. Based on the rate of unlearning success, tuning FFN layers (*e.g.*, FFN, QVFFN) is more receptive to targeted knowledge removal compared to tuning attention layers (*e.g.*, QV, QKVO).

paper (Hu et al., 2022) indicates that applying LoRA to Q and V yields superior performance on downstream tasks, our experiments indicate that using LoRA on Q and V only is insufficient to meet the unlearning criteria within our timeframe of 20 epochs. Notably, when LoRA is applied to FFNs, we observe significant increase in rate of successful unlearning. Furthermore, our LoKU integrating FILA with IHL, achieves the best post-unlearning performance across all LoRA target module combinations.

**Cost-efficiency of the proposed method.** Our compute-cost vs. performance comparisons in Figure 2 show that, while vanilla LoRA allows significant reduction in unlearning costs (*i.e.*, FLOPs) by freezing the majority of parameters, it incurs substantial performance losses compared to full-parameter unlearning due to excessive stability originating from its low-rankness. In contrast, combining the proposed IHL with FILA not only achieves the best performance but also leverages the cost advantages of LoRA.

## 4.3 TASK OF FICTITIOUS UNLEARNING

**Experimental Setup.** The Task of Fictitious Unlearning (TOFU) benchmark (Maini et al., 2024) is a synthetic dataset containing 20 question-answer pairs for each of 200 fictitious author profiles generated by GPT-4. The TOFU evaluation pipeline first finetunes a pretrained LLM on all QA pairs. Given this finetuned LLM that serves as our base model, our task is to unlearn all information regarding 1%, 5%, or 10% of the authors from the model. Note that we can obtain reference models finetuned only on the retain set (QA-pairs on 99%, 95%, or 90% of authors), with which we evaluate the **Forget Quality** of unlearned models by measuring the $p$-value from a Kolmogorov-Smirnov test. A high $p$-value indicates high distributional similarity between the unlearned model and the reference model, thus implying strong forgetting. To evaluate how well the model retains other information outside the forget set, we measure the **Model Utility** as the aggregation of the probability of correct answers, ROUGE-L scores, and Truth Ratio of correct answers vs. incorrect answers on questions from three datasets pertaining to the retain set of fictitious authors, real authors, and world facts.

Following the original paper of TOFU, we prepare two base models by finetuning Phi-1.5B and Llama2-7B on TOFU for 5 epochs with learning rates 2e-5 and 1e-5, respectively. We then unlearn with various methods using LoRA adapters of rank 4, 8, 16, or 32. While we mainly compare our methods against GD, we also compare IHL and FILA against existing unlearning methods such as KL (Maini et al., 2024), DPO (Rafailov et al., 2024), and NPO (Zhang et al., 2024), results from which can be found in Appendix D. For unlearning, we use a learning rate of 2e-4 if our base model is from Phi-1.5B and 1e-4 for Llama2-7B. All training procedures run 5 epochs with an effective batch size of 32 using the AdamW optimizer (Loshchilov & Hutter, 2019).

**Results.** Figure 5 shows the model utility vs. forget quality curves from unlearning three differently-sized TOFU forget sets from Phi-1.5B and Llama2-7B models. Comparing results among different forget set sizes, we first observe that forgetting 1% of author profiles is fairly straightforward, as all curves quickly approach the reference model with a single epoch, with increasing the LoRA rank leading to incremental improvements in performance. On the other hand, when unlearning a larger set of profiles (*i.e.*, 5% or 10%), we see that both GA and GD quickly degrades model utility.

With regards to our proposed method, we find that replacing the NCE loss in GD with our IHL better retains model utility across all LoRA ranks and forget set sizes, as curves are more aligned straight-up towards the reference point with negligible shift in model utility. This stability comes at the cost of unlearning efficiency, however, as randomly initialized LoRA weights are unable to effectively represent weight changes required to decrease IHL. Nonetheless, our LoKU initializing

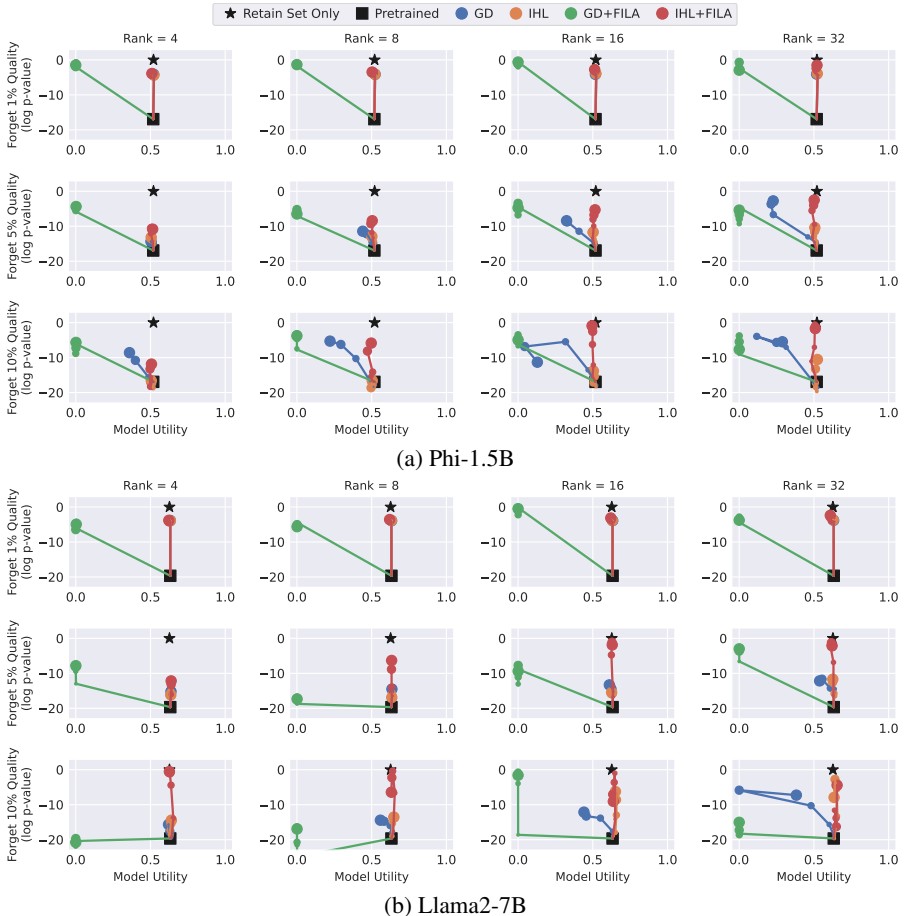

Figure 5: TOFU results using Phi-1.5B and Llama2-7B models. Each row corresponds to unlearning a different forget set (1%, 5%, or 10%), and each column uses a distinct LoRA rank between 4 and 32. The relative size of markers represent the number of epochs. Ideally, the unlearning curves should start from the pretrained model (■) and approach towards the reference model tuned on the retain set only (★) as unlearning progresses. Both GD and GD+FILA suffers from significant loss of model utility due to using GA for unlearning. Replacing GA with IHL largely retains model utility, then our LoKU initializing LoRA adapters with FILA significantly boosts the unlearning efficiency of IHL.

LoRA adapters with FILA largely alleviates this issue and significantly enhances unlearning efficiency of IHL by focusing gradient updates on parameters important to generating $\mathcal{D}_f$.

Interestingly, we find the prior weight assignment via FILA can lead to excessive unlearning in some cases (*e.g.*, unlearning 10% forget set with ranks 8 or 16 on Llama2-7B), with model updates reducing the forget quality after reaching the upper bound at zero. This behavior resembles the *Streisand effect* as unlearning gradients beyond a certain point in optimization unintentionally renders $\mathcal{D}_f$ more noticeable within the model (Golatkar et al., 2020). As reference models are not available for measuring forget quality in real-world scenarios, finding the optimal point at which to stop unlearning to prevent this effect as well as designing a robust evaluation metric that does not depend upon oracle models would be interesting directions, which we leave as future work.

## 5 CONCLUDING REMARKS

In this paper, we address limitations of Gradient Ascent (GA), a widely used method for LLM unlearning, and introduce a novel Inverted Hinge Loss (IHL) to replace the negative cross-entropy loss in GA and resolve issues with dispersed gradients and unboundedness. We also propose Fisher-weighted initialization for low-rank adaptation (FILA) that pre-assigns weights relatively important to generating unwanted information as means to facilitate efficient LLM unlearning with LoRA. Experiments on the Training Data Extraction Challenge dataset with GPT-Neo models along with the TOFU benchmark using Phi-1.5B and Llama2-7B models show that our proposed Low-rank Knowledge Unlearning (LoKU) enables faster and more stable LoRA-based LLM unlearning, significantly outperforming existing baselines in computational efficiency as well as post-unlearning performance.

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

## A    DERIVATIVE ANALYSIS FOR THE INVERTED HINGE LOSS FUNCTION

The function $p_\theta(x_t|x_{<t})$ represents a probability distribution that indicates the likelihood of $x_t$ taking a specific token $x_t$ given the previous tokes $x_{<t}$. This probability is expressed using the softmax function: $p_\theta(x_t|x_{<t}) = \exp(y_t^{(x_t)})/\sum_{v=1}^{V} \exp(y_t^{(v)})$, where $y_t^{(v)}$ denotes the score for the $v$-th token in the vocabulary. To differentiate this function with respect to $y_t^{(x_t)}$, we rewrite $p_\theta(x_t|x_{<t}) = \exp(y_t^{(x_t)})/Z$ where $Z = \sum_{v=1}^{V} \exp(y_t^{(v)})$ is the normalization constant.

We differentiate this function with respect to $y_t^{(k)}$ considering two cases: 1) $k = x_t$ and 2) $k \neq x_t$. For the first case, we can get the following by using the chain rule:

$$\frac{\partial p_\theta(x_t|x_{<t})}{\partial y_t^{(x_t)}} = \frac{\partial}{\partial y_t^{(x_t)}} \left( \frac{\exp(y_t^{(x_t)})}{Z} \right) = \frac{1}{Z} \frac{\partial \exp(y_t^{(x_t)})}{\partial y_t^{(x_t)}} - \frac{\exp(y_t^{(x_t)})}{Z^2} \frac{\partial Z}{\partial y_t^{(x_t)}}$$

Here, $\frac{\partial \exp(y_t^{(x_t)})}{\partial y_t^{(x_t)}} = \exp(y_t^{(x_t)})$ and $\frac{\partial Z}{\partial y_t^{(x_t)}} = \exp(y_t^{(x_t)})$. Therefore, it becomes:

$$\frac{\partial p_\theta(x_t|x_{<t})}{\partial y_t^{(x_t)}} = \frac{\exp(y_t^{(x_t)})}{Z} - \frac{\exp(y_t^{(x_t)})^2}{Z^2} = p_\theta(x_t|x_{<t}) - p_\theta(x_t|x_{<t})^2 = p_\theta(x_t|x_{<t})(1 - p_\theta(x_t|x_{<t}))$$

For the second case, using the chain rule again, we get:

$$\frac{\partial p_\theta(x_t|x_{<t})}{\partial y_t^{(k)}} = \frac{\partial}{\partial y_t^{(k)}} \left( \frac{\exp(y_t^{(x_t)})}{Z} \right) = -\frac{\exp(y_t^{(x_t)})}{Z^2} \frac{\partial Z}{\partial y_t^{(k)}}$$

where $\frac{\partial Z}{\partial y_t^{(k)}} = \exp(y_t^{(k)})$. Therefore,

$$\frac{\partial p_\theta(x_t|x_{<t})}{\partial y_t^{(k)}} = -\frac{\exp(y_t^{(x_t)}) \exp(y_t^{(k)})}{Z^2} = -p_\theta(x_t|x_{<t}) \cdot p_\theta(k|x_{<t})$$

Thus, we can summarize them as below:

$$\frac{\partial p_\theta(x_t|x_{<t})}{\partial y_t^{(v)}} = \begin{cases} p_\theta(x_t|x_{<t})(1 - p_\theta(x_t|x_{<t})) & \text{if } v = x_t \\ -p_\theta(x_t|x_{<t}) \cdot p_\theta(v|x_{<t}) & \text{if } v \neq x_t \end{cases}$$

Based on the derivative of $p_\theta(x_t|x_{<t})$ above , we can calculate the derivative of $\mathcal{L}_{\text{IHL}}$. Firstly, for convenience, we define $p_t = p_\theta(x_t|x_{<t})$ and $\hat{p}_t = \max_{v \neq x_t}(p_\theta(v|x_{<t}))$. The loss function can be rewritten as:

$$\mathcal{L}_{\text{IHL}}(\boldsymbol{x}) = 1 + p_t - \hat{p}_t$$

To calculate the derivative of $\mathcal{L}_{\text{IHL}}$, we need to consider three cases: 1) when $v = x_t$, 2) when $v = v^\star$ where $v^\star = \arg\max_{v \neq x_t} p_\theta(v|x_{<t})$, 3) when $v \neq x_t$ and $v \neq v^\star$. Using the derivative of $p_\theta(x_t|x_{<t})$ mentioned earlier, the derivative of $\mathcal{L}_{\text{IHL}}$ with respect to $y_t^{(v)}$ is as follows:

$$\begin{aligned} \frac{\partial \mathcal{L}_{\text{IHL}}}{\partial y_t^{(x_t)}} &= \frac{\partial}{\partial y_t^{(x_t)}} (1 + p_\theta(x_t|x_{<t}) - p_\theta(v^\star|x_{<t})) \\ &= p_\theta(x_t|x_{<t})(1 - p_\theta(x_t|x_{<t})) + p_\theta(x_t|x_{<t}) \cdot p_\theta(v^\star|x_{<t}) \\ &= p_\theta(x_t|x_{<t})(1 - p_\theta(x_t|x_{<t}) + p_\theta(v^\star|x_{<t})) \end{aligned}$$

$$\frac{\partial \mathcal{L}_{\text{IHL}}}{\partial y_t^{(v^\star)}} = \frac{\partial}{\partial y_t^{(v^\star)}} \left(1 + p_\theta(x_t|x_{<t}) - p_\theta(v^\star|x_{<t})\right)$$
$$= -p_\theta(x_t|x_{<t}) \cdot p_\theta(v^\star|x_{<t}) - p_\theta(v^\star|x_{<t})(1 - p_\theta(v^\star|x_{<t}))$$
$$= -p_\theta(v^\star|x_{<t}) \left(1 - p_\theta(v^\star|x_{<t}) + p_\theta(x_t|x_{<t})\right)$$

$$\frac{\partial \mathcal{L}_{\text{IHL}}}{\partial y_t^{(v)}} = \frac{\partial}{\partial y_t^{(v)}} \left(1 + p_\theta(x_t|x_{<t}) - p_\theta(v^\star|x_{<t})\right)$$
$$= -p_\theta(x_t|x_{<t}) \cdot p_\theta(v|x_{<t}) + p_\theta(v^\star|x_{<t}) \cdot p_\theta(v|x_{<t})$$
$$= p_\theta(v|x_{<t}) \left(p_\theta(v^\star|x_{<t}) - p_\theta(x_t|x_{<t})\right)$$

In summary, the derivatives of the loss function $\mathcal{L}_{\text{IHL}}$ with respect to $y_t^{(v)}$ for the three cases are:

$$\frac{\partial \mathcal{L}_{\text{IHL}}(\boldsymbol{x})}{\partial y_t^{(v)}} = \begin{cases} p_\theta(x_t|x_{<t})(p_\theta(v^\star|x_{<t}) - p_\theta(x_t|x_{<t}) + 1) & \text{if } v = x_t \\ p_\theta(v^\star|x_{<t})(p_\theta(v^\star|x_{<t}) - p_\theta(x_t|x_{<t}) - 1) & \text{if } v = v^\star \\ p_\theta(v|x_{<t})(p_\theta(v^\star|x_{<t}) - p_\theta(x_t|x_{<t})) & \text{if } v \neq x_t \text{ and } v \neq v^\star, \end{cases}$$

## B  EVALUATION METRICS

**How to measure success of unlearning?** Following previous work Jang et al. (2023); Tirumala et al. (2022), we empirically measure the success of unlearning using two metrics, Extraction Likelihood (EL) and Memorization Accuracy (MA), which we briefly discuss below.

After unlearning each sequence $\boldsymbol{x} = (x_1, \ldots, x_T) \in \mathcal{D}_f$, the Extraction Likelihood (EL) is measured as the $n$-gram overlap between the ground truth sequence $\boldsymbol{x}$ and the output of the model after unlearning.

$$\text{OVERLAP}_n(\boldsymbol{a}, \boldsymbol{b}) = \frac{\sum_{\boldsymbol{c} \in n\text{-GRAM}(\boldsymbol{a})} \mathbb{1}\{\boldsymbol{c} \in n\text{-GRAM}(\boldsymbol{b})\}}{|n\text{-GRAM}(\boldsymbol{a})|} \tag{4}$$

$$\text{EL}_n(\boldsymbol{x}) = \frac{\sum_{t=1}^{T-n} \text{OVERLAP}_n\left(f_\theta(x_{<t}), x_{\geq t}\right)}{T - n} \tag{5}$$

The Memorization Accuracy (MA) measures the token-wise memorization of the LM $p_\theta$.

$$\text{MA}(\boldsymbol{x}) = \frac{\sum_{t=1}^{T} \mathbb{1}\{\arg\max_x p_\theta(x|x_{<t}) = x_t\}}{T - 1} \tag{6}$$

Given these two metrics, we flag successful unlearning when the average EL and MA on $\mathcal{D}_f$ goes below the EL and MA values measured on the validation set unseen during training. In our experiments we measure EL with 10-grams, which results in the following early stopping criterion.

$$\frac{1}{\mathcal{D}_f} \sum_{\boldsymbol{x} \in \mathcal{D}_f} \text{EL}_{10}(\boldsymbol{x}) \leq \frac{1}{\mathcal{D}_{\text{val}}} \sum_{\boldsymbol{x} \in \mathcal{D}_{\text{val}}} \text{EL}_{10}(\boldsymbol{x}) \quad \text{and} \quad \frac{1}{\mathcal{D}_f} \sum_{\boldsymbol{x} \in \mathcal{D}_f} \text{MA}(\boldsymbol{x}) \leq \frac{1}{\mathcal{D}_{\text{val}}} \sum_{\boldsymbol{x} \in \mathcal{D}_{\text{val}}} \text{MA}(\boldsymbol{x})$$

## C  ADDITIONAL DETAILS ON EXPERIMENTAL SETTING

**Experiemtal Settings**  All experiments were conducted on a remote server equipped with NVIDIA A100 40GB Tensor Core GPUs.

**Datasets for Evaluation in the TDEC**  To evaluate reasoning capabilities, we utilize nine different classification datasets: LAMBADA (Paperno et al., 2016), Hellaswag (Zellers et al., 2019),

Winogrande (Sakaguchi et al., 2021), COPA (Gordon et al., 2012), ARC-Easy (Clark et al., 2018), ARC-Challenge (Clark et al., 2018), PiQA (Bisk et al., 2020), MathQA (Amini et al., 2019), and PubmedQA (Jin et al., 2019). To assess generative performance, we employ Blended Skill Talk (Smith et al., 2020), Empathetic Dialogues (Rashkin et al., 2019), Wizard of Internet (Komeili et al., 2022), and Wizard of Wikipedia (Dinan et al., 2019).

**Details of metrics of TOFU** We evaluate the **Forget Quality** of unlearned models by measuring the $p$-value from the Kolmogorov-Smirnov test that compares the empirical distribution of our unlearned model to that of the reference model. To evaluate how well the model retains other information outside the forget set, we measure the **Model Utility** as the aggregated model performance on the retain set of remaining fictitious author profiles, and two held-out sets consisted of QA-pairs regarding real author profiles and other world facts.

## D    ADDITIONAL EXPERIMENTAL RESULTS

### D.1    CONTINUAL UNLEARNING

Because of the importance of continual unlearning (or sequential unlearning) in real-world applications, previous studies have underscored its relevance through a sequence of unlearning tasks (Cha et al., 2024; Jang et al., 2023). Building on them, we conduct continual unlearning experiments involving four tasks. Figure 6 of the Appendix shows that IHL consistently outperforms GD across all metrics. Notably, the proposed IHL demonstrates significantly enhanced performance on the four Dialogue and Pile datasets. Finally, we confirm that the combination of IHL and FLoRA achieves more robust and cose-efficient continual unlearning, as evidenced by the experimental results for Reasoning, Dialogue, and Pile, while utilizing only about 1.6% of the total parameters.

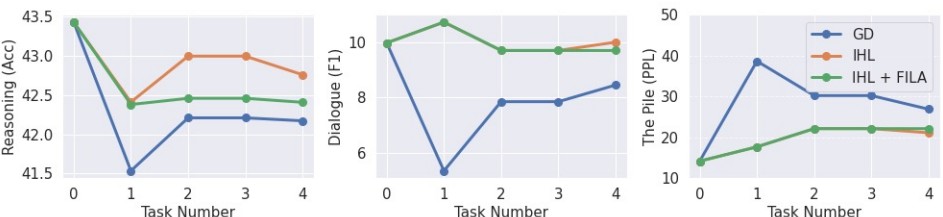

Figure 6: Experimental results of continual unlearning. Each task consists of 32 disjoint sequences sampled from the TDEC dataset, leading to a total of 128 sequences to unlearn. For these experiments, we use the pretrained GPT-Neo 125M model. The experimental setup for unlearning and the forgetting criteria are configured as in the previous TDEC experiments. Task 0 refers to the result before unlearning.

### D.2    TOFU RESULTS WITH ADDITIONAL BASELINES

We compare our IHL and FILA methods against three additional existing unlearning baselines: KL (Maini et al., 2024) uses GA to forget samples in $\mathcal{D}_f$ while minimizing the Kullback-Leibler (KL) divergence between representations of retain samples in $\mathcal{D}_r$ output by the unlearned model and those from the pretrained base model. Instead of GA, Direct Preference Optimization (DPO; Rafailov et al. (2024)) performs unlearning by training the model to output variants of "I don't know" when given a sample in $\mathcal{D}_f$. NPO (Zhang et al., 2024) is an approach similar to GA, but with adaptive weighting on the gradients such that it alleviates the divergent behavior of GA. Note that both DPO and NPO are regularized by the LM-loss on $\mathcal{D}_r$. We use the same hyperparameterization (*e.g.,* learning rate and effective batch size) as in our main results. Figure 7 shows the results. We find that all three baselines lead to significant decrease in model utility, while IHL+FILA shows negligible change.

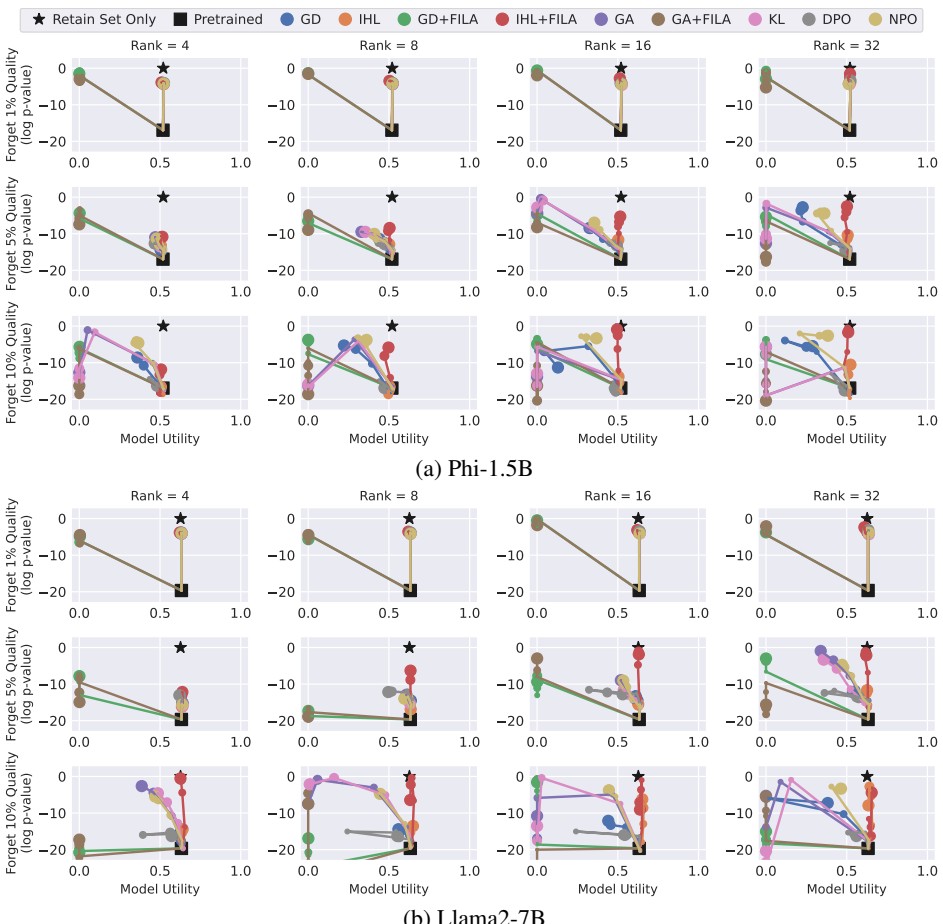

Figure 7: TOFU results using Phi-1.5B and Llama2-7B models. Each row corresponds to unlearning a different forget set (1%, 5%, or 10%), and each column uses a distinct LoRA rank between 4 and 32. The relative size of markers represent the number of epochs. Ideally, the unlearning curves should start from the pretrained model (■) and approach towards the reference model tuned on the retain set only (★) as unlearning progresses. Our method IHL+FILA outperforms existing KL- and preference optimization-based unlearning approaches in both model utility and forget quality.

| Method | Forget Quality | | | | Model Utility | | | | | | | | | |
| --- | --- | --- | --- | --- | --- | --- | --- | --- | --- | --- | --- | --- | --- | --- |
| | Forget Set | | | FQ (↑) | Retain Set | | | Real Authors | | | Real World | | | MU (↑) |
| | Rouge-L | Prob. | Truth Ratio | | Rouge-L | Prob. | Truth Ratio | Rouge-L | Prob. | Truth Ratio | Rouge-L | Prob. | Truth Ratio | |
| Original | 0.93 | 0.93 | 0.48 | 1.15e-17 | 0.92 | 0.92 | 0.48 | 0.41 | 0.37 | 0.45 | 0.75 | 0.41 | 0.50 | 0.52 |
| Retain90 | 0.43 | 0.14 | 0.63 | 1.00e+00 | 0.91 | 0.92 | 0.48 | 0.43 | 0.38 | 0.45 | 0.76 | 0.41 | 0.49 | 0.52 |
| **TOFU Forget01** | | | | | | | | | | | | | | |
| KL | 0.96 | 0.92 | 0.48 | 7.37e-05 | 0.92 | 0.92 | 0.48 | 0.43 | 0.38 | 0.45 | 0.76 | 0.41 | 0.50 | 0.52 |
| DPO | 0.96 | 0.93 | 0.48 | 8.87e-05 | 0.92 | 0.92 | 0.48 | 0.44 | 0.38 | 0.45 | 0.75 | 0.41 | 0.50 | 0.52 |
| NPO | 0.96 | 0.92 | 0.48 | 6.11e-05 | 0.92 | 0.92 | 0.48 | 0.43 | 0.37 | 0.45 | 0.76 | 0.41 | 0.50 | 0.52 |
| GA | 0.96 | 0.92 | 0.48 | 6.11e-05 | 0.92 | 0.92 | 0.48 | 0.43 | 0.38 | 0.45 | 0.76 | 0.41 | 0.50 | 0.52 |
| GA+FILA | 0.04 | 0.00 | 0.76 | 1.07e-03 | 0.06 | 0.00 | 0.21 | 0.01 | 0.27 | 0.29 | 0.02 | 0.29 | 0.30 | 0.00 |
| GD | 0.96 | 0.93 | 0.48 | 7.37e-05 | 0.92 | 0.92 | 0.48 | 0.42 | 0.38 | 0.45 | 0.76 | 0.41 | 0.50 | 0.52 |
| IHL | 0.96 | 0.93 | 0.48 | 4.17e-05 | 0.92 | 0.92 | 0.48 | 0.43 | 0.38 | 0.45 | 0.75 | 0.41 | 0.50 | 0.52 |
| GD+FILA | 0.03 | 0.00 | 0.69 | 3.24e-02 | 0.06 | 0.00 | 0.20 | 0.00 | 0.26 | 0.31 | 0.03 | 0.27 | 0.31 | 0.00 |
| IHL+FILA | 0.50 | 0.52 | 0.49 | 1.28e-04 | 0.83 | 0.89 | 0.49 | 0.37 | 0.38 | 0.45 | 0.73 | 0.41 | 0.50 | 0.51 |
| **TOFU Forget05** | | | | | | | | | | | | | | |
| KL | 0.62 | 0.49 | 0.51 | 2.90e-13 | 0.65 | 0.51 | 0.46 | 0.48 | 0.36 | 0.43 | 0.80 | 0.39 | 0.47 | 0.48 |
| DPO | 0.43 | 0.82 | 0.51 | 2.17e-13 | 0.55 | 0.82 | 0.45 | 0.34 | 0.35 | 0.42 | 0.72 | 0.41 | 0.50 | 0.47 |
| NPO | 0.62 | 0.48 | 0.51 | 4.87e-12 | 0.64 | 0.51 | 0.45 | 0.50 | 0.36 | 0.43 | 0.80 | 0.39 | 0.47 | 0.48 |
| GA | 0.61 | 0.46 | 0.51 | 1.10e-11 | 0.63 | 0.48 | 0.45 | 0.46 | 0.36 | 0.43 | 0.80 | 0.39 | 0.46 | 0.47 |
| GA+FILA | 0.09 | 0.00 | 0.73 | 3.69e-08 | 0.10 | 0.00 | 0.20 | 0.00 | 0.25 | 0.28 | 0.03 | 0.24 | 0.25 | 0.00 |
| GD | 0.70 | 0.80 | 0.47 | 4.33e-15 | 0.79 | 0.88 | 0.48 | 0.37 | 0.38 | 0.45 | 0.72 | 0.41 | 0.50 | 0.50 |
| IHL | 0.71 | 0.82 | 0.48 | 6.68e-14 | 0.83 | 0.90 | 0.48 | 0.37 | 0.38 | 0.45 | 0.73 | 0.41 | 0.49 | 0.50 |
| GD+FILA | 0.12 | 0.00 | 0.72 | 4.33e-05 | 0.13 | 0.00 | 0.18 | 0.01 | 0.28 | 0.36 | 0.02 | 0.29 | 0.32 | 0.00 |
| IHL+FILA | 0.45 | 0.37 | 0.50 | 1.44e-11 | 0.79 | 0.86 | 0.48 | 0.43 | 0.37 | 0.46 | 0.75 | 0.41 | 0.50 | 0.51 |
| **TOFU Forget10** | | | | | | | | | | | | | | |
| KL | 0.01 | 0.00 | 0.77 | 7.38e-15 | 0.01 | 0.00 | 0.16 | 0.00 | 0.26 | 0.24 | 0.00 | 0.27 | 0.25 | 0.00 |
| DPO | 0.41 | 0.87 | 0.49 | 5.10e-17 | 0.67 | 0.88 | 0.47 | 0.33 | 0.36 | 0.43 | 0.73 | 0.41 | 0.49 | 0.48 |
| NPO | 0.45 | 0.21 | 0.61 | 2.56e-05 | 0.45 | 0.20 | 0.38 | 0.35 | 0.34 | 0.39 | 0.71 | 0.37 | 0.43 | 0.37 |
| GA | 0.01 | 0.00 | 0.76 | 2.06e-13 | 0.01 | 0.00 | 0.15 | 0.00 | 0.26 | 0.24 | 0.00 | 0.26 | 0.24 | 0.00 |
| GA+FILA | 0.00 | 0.00 | 0.35 | 5.10e-17 | 0.00 | 0.00 | 0.25 | 0.00 | 0.25 | 0.38 | 0.00 | 0.23 | 0.32 | 0.00 |
| GD | 0.37 | 0.19 | 0.53 | 2.55e-09 | 0.41 | 0.29 | 0.44 | 0.19 | 0.36 | 0.44 | 0.60 | 0.38 | 0.46 | 0.36 |
| IHL | 0.53 | 0.67 | 0.49 | 2.43e-17 | 0.76 | 0.86 | 0.49 | 0.39 | 0.38 | 0.45 | 0.71 | 0.41 | 0.50 | 0.51 |
| GD+FILA | 0.12 | 0.00 | 0.65 | 2.17e-06 | 0.11 | 0.00 | 0.23 | 0.00 | 0.30 | 0.30 | 0.03 | 0.26 | 0.28 | 0.00 |
| IHL+FILA | 0.26 | 0.19 | 0.49 | 1.39e-12 | 0.75 | 0.85 | 0.50 | 0.36 | 0.39 | 0.49 | 0.67 | 0.41 | 0.51 | 0.51 |

Table 2: Detailed TOFU Results using the Phi-1.5B model with LoRA rank = 4.

| Method | Forget Quality | | | | Model Utility | | | | | | | | | |
| --- | --- | --- | --- | --- | --- | --- | --- | --- | --- | --- | --- | --- | --- | --- |
| | Forget Set | | | FQ (↑) | Retain Set | | | Real Authors | | | Real World | | | MU (↑) |
| | Rouge-L | Prob. | Truth Ratio | | Rouge-L | Prob. | Truth Ratio | Rouge-L | Prob. | Truth Ratio | Rouge-L | Prob. | Truth Ratio | |
| Original | 0.93 | 0.93 | 0.48 | 1.15e-17 | 0.92 | 0.92 | 0.48 | 0.41 | 0.37 | 0.45 | 0.75 | 0.41 | 0.50 | 0.52 |
| Retain90 | 0.43 | 0.14 | 0.63 | 1.00e+00 | 0.91 | 0.92 | 0.48 | 0.43 | 0.38 | 0.45 | 0.76 | 0.41 | 0.49 | 0.52 |
| **TOFU Forget01** | | | | | | | | | | | | | | |
| KL | 0.91 | 0.91 | 0.48 | 6.11e-05 | 0.92 | 0.92 | 0.48 | 0.43 | 0.37 | 0.45 | 0.77 | 0.41 | 0.50 | 0.52 |
| DPO | 0.96 | 0.93 | 0.49 | 8.87e-05 | 0.92 | 0.92 | 0.48 | 0.43 | 0.38 | 0.45 | 0.76 | 0.41 | 0.50 | 0.52 |
| NPO | 0.92 | 0.91 | 0.48 | 6.11e-05 | 0.91 | 0.92 | 0.48 | 0.43 | 0.37 | 0.45 | 0.76 | 0.41 | 0.50 | 0.52 |
| GA | 0.92 | 0.91 | 0.48 | 4.17e-05 | 0.92 | 0.92 | 0.48 | 0.43 | 0.37 | 0.45 | 0.77 | 0.41 | 0.50 | 0.52 |
| GA+FILA | 0.00 | 0.00 | 0.65 | 2.72e-02 | 0.01 | 0.00 | 0.22 | 0.00 | 0.23 | 0.32 | 0.00 | 0.29 | 0.34 | 0.00 |
| GD | 0.93 | 0.92 | 0.48 | 7.37e-05 | 0.92 | 0.92 | 0.48 | 0.43 | 0.38 | 0.45 | 0.76 | 0.41 | 0.50 | 0.52 |
| IHL | 0.94 | 0.92 | 0.48 | 7.37e-05 | 0.92 | 0.92 | 0.48 | 0.43 | 0.38 | 0.45 | 0.75 | 0.41 | 0.50 | 0.52 |
| GD+FILA | 0.01 | 0.00 | 0.65 | 4.55e-02 | 0.01 | 0.00 | 0.24 | 0.00 | 0.23 | 0.32 | 0.02 | 0.29 | 0.36 | 0.00 |
| IHL+FILA | 0.47 | 0.33 | 0.51 | 3.37e-04 | 0.80 | 0.86 | 0.49 | 0.34 | 0.38 | 0.46 | 0.73 | 0.42 | 0.51 | 0.50 |
| **TOFU Forget05** | | | | | | | | | | | | | | |
| KL | 0.46 | 0.14 | 0.55 | 3.26e-10 | 0.45 | 0.15 | 0.44 | 0.38 | 0.34 | 0.42 | 0.71 | 0.36 | 0.43 | 0.35 |
| DPO | 0.44 | 0.84 | 0.50 | 1.21e-13 | 0.64 | 0.85 | 0.46 | 0.34 | 0.36 | 0.42 | 0.72 | 0.41 | 0.50 | 0.48 |
| NPO | 0.51 | 0.24 | 0.54 | 9.13e-11 | 0.50 | 0.25 | 0.43 | 0.48 | 0.34 | 0.42 | 0.78 | 0.37 | 0.44 | 0.41 |
| GA | 0.45 | 0.12 | 0.55 | 3.26e-10 | 0.45 | 0.13 | 0.43 | 0.33 | 0.34 | 0.42 | 0.69 | 0.36 | 0.43 | 0.33 |
| GA+FILA | 0.02 | 0.00 | 0.54 | 1.12e-09 | 0.02 | 0.00 | 0.22 | 0.00 | 0.23 | 0.33 | 0.00 | 0.25 | 0.30 | 0.00 |
| GD | 0.47 | 0.49 | 0.49 | 3.70e-12 | 0.53 | 0.66 | 0.47 | 0.26 | 0.37 | 0.44 | 0.69 | 0.40 | 0.48 | 0.44 |
| IHL | 0.62 | 0.74 | 0.48 | 1.21e-13 | 0.81 | 0.89 | 0.48 | 0.37 | 0.37 | 0.45 | 0.72 | 0.41 | 0.49 | 0.50 |
| GD+FILA | 0.02 | 0.00 | 0.74 | 3.16e-07 | 0.02 | 0.00 | 0.16 | 0.00 | 0.26 | 0.32 | 0.02 | 0.29 | 0.33 | 0.00 |
| IHL+FILA | 0.41 | 0.25 | 0.52 | 3.73e-09 | 0.77 | 0.86 | 0.48 | 0.38 | 0.37 | 0.47 | 0.71 | 0.41 | 0.50 | 0.50 |
| **TOFU Forget10** | | | | | | | | | | | | | | |
| KL | 0.01 | 0.00 | 0.76 | 1.06e-16 | 0.01 | 0.00 | 0.13 | 0.00 | 0.27 | 0.34 | 0.00 | 0.26 | 0.27 | 0.00 |
| DPO | 0.39 | 0.87 | 0.49 | 1.15e-17 | 0.72 | 0.89 | 0.47 | 0.28 | 0.37 | 0.43 | 0.73 | 0.41 | 0.49 | 0.47 |
| NPO | 0.46 | 0.24 | 0.63 | 1.66e-04 | 0.46 | 0.24 | 0.36 | 0.27 | 0.33 | 0.38 | 0.71 | 0.37 | 0.43 | 0.36 |
| GA | 0.01 | 0.00 | 0.76 | 5.10e-17 | 0.01 | 0.00 | 0.13 | 0.00 | 0.27 | 0.34 | 0.00 | 0.26 | 0.27 | 0.00 |
| GA+FILA | 0.00 | 0.00 | 0.39 | 2.43e-19 | 0.00 | 0.00 | 0.29 | 0.00 | 0.24 | 0.36 | 0.00 | 0.26 | 0.34 | 0.00 |
| GD | 0.23 | 0.04 | 0.55 | 5.07e-06 | 0.27 | 0.13 | 0.43 | 0.07 | 0.35 | 0.45 | 0.44 | 0.38 | 0.45 | 0.22 |
| IHL | 0.46 | 0.55 | 0.49 | 2.43e-17 | 0.77 | 0.85 | 0.49 | 0.38 | 0.38 | 0.45 | 0.69 | 0.41 | 0.50 | 0.50 |
| GD+FILA | 0.10 | 0.00 | 0.60 | 1.66e-04 | 0.10 | 0.00 | 0.19 | 0.00 | 0.29 | 0.41 | 0.04 | 0.27 | 0.32 | 0.00 |
| IHL+FILA | 0.18 | 0.09 | 0.52 | 1.40e-06 | 0.73 | 0.84 | 0.49 | 0.30 | 0.41 | 0.51 | 0.66 | 0.43 | 0.52 | 0.50 |

Table 3: Detailed TOFU Results using the Phi-1.5B model with LoRA rank = 8.

| Method | Forget Quality | | | | Model Utility | | | | | | | | | |
|---|---|---|---|---|---|---|---|---|---|---|---|---|---|---|
| | Forget Set | | | FQ (↑) | Retain Set | | | Real Authors | | | Real World | | | MU (↑) |
| | Rouge-L | Prob. | Truth Ratio | | Rouge-L | Prob. | Truth Ratio | Rouge-L | Prob. | Truth Ratio | Rouge-L | Prob. | Truth Ratio | |
| Original | 0.93 | 0.93 | 0.48 | 1.15e-17 | 0.92 | 0.92 | 0.48 | 0.41 | 0.37 | 0.45 | 0.75 | 0.41 | 0.50 | 0.52 |
| Retain90 | 0.43 | 0.14 | 0.63 | 1.00e+00 | 0.91 | 0.92 | 0.48 | 0.43 | 0.38 | 0.45 | 0.76 | 0.41 | 0.49 | 0.52 |
| **TOFU Forget01** | | | | | | | | | | | | | | |
| KL | 0.84 | 0.86 | 0.48 | 2.83e-05 | 0.91 | 0.92 | 0.48 | 0.46 | 0.37 | 0.45 | 0.75 | 0.41 | 0.49 | 0.53 |
| DPO | 0.96 | 0.93 | 0.49 | 7.37e-05 | 0.91 | 0.92 | 0.48 | 0.42 | 0.38 | 0.45 | 0.76 | 0.42 | 0.51 | 0.52 |
| NPO | 0.82 | 0.86 | 0.48 | 4.17e-05 | 0.91 | 0.92 | 0.48 | 0.44 | 0.37 | 0.45 | 0.76 | 0.41 | 0.49 | 0.52 |
| GA | 0.84 | 0.86 | 0.48 | 6.11e-05 | 0.91 | 0.92 | 0.48 | 0.44 | 0.37 | 0.45 | 0.75 | 0.41 | 0.49 | 0.52 |
| GA+FILA | 0.03 | 0.00 | 0.64 | 1.14e-02 | 0.01 | 0.00 | 0.19 | 0.01 | 0.22 | 0.27 | 0.01 | 0.29 | 0.29 | 0.00 |
| GD | 0.88 | 0.88 | 0.48 | 7.37e-05 | 0.92 | 0.92 | 0.49 | 0.40 | 0.38 | 0.45 | 0.76 | 0.41 | 0.50 | 0.52 |
| IHL | 0.88 | 0.89 | 0.48 | 1.28e-04 | 0.91 | 0.92 | 0.49 | 0.42 | 0.38 | 0.45 | 0.76 | 0.41 | 0.50 | 0.52 |
| GD+FILA | 0.03 | 0.00 | 0.60 | 2.44e-01 | 0.02 | 0.00 | 0.19 | 0.00 | 0.22 | 0.27 | 0.01 | 0.30 | 0.36 | 0.00 |
| IHL+FILA | 0.44 | 0.19 | 0.55 | 1.70e-03 | 0.75 | 0.84 | 0.49 | 0.37 | 0.39 | 0.47 | 0.72 | 0.43 | 0.53 | 0.51 |
| **TOFU Forget05** | | | | | | | | | | | | | | |
| KL | 0.21 | 0.00 | 0.69 | 1.53e-03 | 0.22 | 0.00 | 0.29 | 0.02 | 0.27 | 0.34 | 0.04 | 0.27 | 0.29 | 0.00 |
| DPO | 0.43 | 0.85 | 0.50 | 6.68e-14 | 0.73 | 0.88 | 0.47 | 0.37 | 0.36 | 0.43 | 0.74 | 0.41 | 0.50 | 0.49 |
| NPO | 0.46 | 0.16 | 0.58 | 1.10e-07 | 0.45 | 0.17 | 0.41 | 0.36 | 0.34 | 0.41 | 0.66 | 0.36 | 0.43 | 0.35 |
| GA | 0.21 | 0.00 | 0.71 | 4.33e-05 | 0.21 | 0.00 | 0.26 | 0.01 | 0.26 | 0.32 | 0.04 | 0.26 | 0.27 | 0.00 |
| GA+FILA | 0.02 | 0.00 | 0.46 | 5.96e-09 | 0.02 | 0.00 | 0.29 | 0.00 | 0.25 | 0.34 | 0.00 | 0.27 | 0.38 | 0.00 |
| GD | 0.40 | 0.19 | 0.52 | 3.73e-09 | 0.43 | 0.39 | 0.45 | 0.12 | 0.35 | 0.41 | 0.53 | 0.37 | 0.45 | 0.32 |
| IHL | 0.52 | 0.59 | 0.49 | 2.12e-12 | 0.79 | 0.87 | 0.48 | 0.38 | 0.38 | 0.45 | 0.71 | 0.41 | 0.49 | 0.50 |
| GD+FILA | 0.08 | 0.00 | 0.69 | 1.81e-05 | 0.07 | 0.00 | 0.17 | 0.01 | 0.25 | 0.33 | 0.05 | 0.27 | 0.27 | 0.00 |
| IHL+FILA | 0.36 | 0.11 | 0.57 | 5.03e-06 | 0.75 | 0.84 | 0.49 | 0.43 | 0.37 | 0.47 | 0.71 | 0.42 | 0.51 | 0.52 |
| **TOFU Forget10** | | | | | | | | | | | | | | |
| KL | 0.01 | 0.00 | 0.70 | 1.07e-13 | 0.01 | 0.00 | 0.14 | 0.00 | 0.29 | 0.41 | 0.00 | 0.25 | 0.35 | 0.00 |
| DPO | 0.32 | 0.86 | 0.48 | 5.40e-18 | 0.76 | 0.90 | 0.48 | 0.32 | 0.37 | 0.43 | 0.72 | 0.41 | 0.49 | 0.48 |
| NPO | 0.45 | 0.25 | 0.65 | 4.69e-04 | 0.45 | 0.27 | 0.35 | 0.30 | 0.33 | 0.37 | 0.69 | 0.37 | 0.42 | 0.37 |
| GA | 0.01 | 0.00 | 0.71 | 1.46e-14 | 0.01 | 0.00 | 0.14 | 0.00 | 0.29 | 0.41 | 0.00 | 0.25 | 0.35 | 0.00 |
| GA+FILA | 0.00 | 0.00 | 0.31 | 5.10e-17 | 0.00 | 0.00 | 0.28 | 0.00 | 0.25 | 0.33 | 0.00 | 0.26 | 0.43 | 0.00 |
| GD | 0.20 | 0.00 | 0.52 | 4.78e-12 | 0.25 | 0.12 | 0.46 | 0.02 | 0.35 | 0.50 | 0.28 | 0.42 | 0.50 | 0.13 |
| IHL | 0.41 | 0.38 | 0.51 | 1.46e-14 | 0.77 | 0.85 | 0.49 | 0.36 | 0.38 | 0.46 | 0.69 | 0.42 | 0.52 | 0.50 |
| GD+FILA | 0.08 | 0.00 | 0.50 | 1.16e-05 | 0.09 | 0.00 | 0.22 | 0.00 | 0.31 | 0.52 | 0.04 | 0.26 | 0.34 | 0.00 |
| IHL+FILA | 0.13 | 0.03 | 0.56 | 1.21e-01 | 0.70 | 0.80 | 0.47 | 0.32 | 0.40 | 0.48 | 0.67 | 0.44 | 0.55 | 0.50 |

Table 4: Detailed TOFU Results using the Phi-1.5B model with LoRA rank = 16.

| Method | Forget Quality | | | | Model Utility | | | | | | | | | |
|---|---|---|---|---|---|---|---|---|---|---|---|---|---|---|
| | Forget Set | | | FQ (↑) | Retain Set | | | Real Authors | | | Real World | | | MU (↑) |
| | Rouge-L | Prob. | Truth Ratio | | Rouge-L | Prob. | Truth Ratio | Rouge-L | Prob. | Truth Ratio | Rouge-L | Prob. | Truth Ratio | |
| Original | 0.93 | 0.93 | 0.48 | 1.15e-17 | 0.92 | 0.92 | 0.48 | 0.41 | 0.37 | 0.45 | 0.75 | 0.41 | 0.50 | 0.52 |
| Retain90 | 0.43 | 0.14 | 0.63 | 1.00e+00 | 0.91 | 0.92 | 0.48 | 0.43 | 0.38 | 0.45 | 0.76 | 0.41 | 0.49 | 0.52 |
| **TOFU Forget01** | | | | | | | | | | | | | | |
| KL | 0.68 | 0.72 | 0.48 | 4.17e-05 | 0.87 | 0.90 | 0.49 | 0.43 | 0.37 | 0.45 | 0.77 | 0.41 | 0.49 | 0.52 |
| DPO | 0.84 | 0.90 | 0.51 | 4.72e-04 | 0.87 | 0.91 | 0.47 | 0.43 | 0.38 | 0.45 | 0.76 | 0.42 | 0.52 | 0.52 |
| NPO | 0.65 | 0.71 | 0.49 | 5.05e-05 | 0.87 | 0.89 | 0.48 | 0.42 | 0.37 | 0.44 | 0.75 | 0.41 | 0.49 | 0.51 |
| GA | 0.67 | 0.72 | 0.49 | 5.56e-05 | 0.87 | 0.89 | 0.48 | 0.42 | 0.37 | 0.45 | 0.75 | 0.41 | 0.49 | 0.51 |
| GA+FILA | 0.03 | 0.00 | 0.78 | 5.55e-06 | 0.02 | 0.00 | 0.16 | 0.00 | 0.26 | 0.27 | 0.01 | 0.26 | 0.28 | 0.00 |
| GD | 0.68 | 0.78 | 0.48 | 8.87e-05 | 0.90 | 0.92 | 0.49 | 0.40 | 0.38 | 0.45 | 0.75 | 0.41 | 0.50 | 0.52 |
| IHL | 0.65 | 0.77 | 0.48 | 1.28e-04 | 0.90 | 0.92 | 0.49 | 0.42 | 0.38 | 0.45 | 0.76 | 0.41 | 0.50 | 0.52 |
| GD+FILA | 0.04 | 0.00 | 0.77 | 1.15e-03 | 0.03 | 0.00 | 0.17 | 0.00 | 0.26 | 0.24 | 0.02 | 0.25 | 0.26 | 0.00 |
| IHL+FILA | 0.37 | 0.09 | 0.61 | 3.06e-02 | 0.71 | 0.82 | 0.49 | 0.43 | 0.39 | 0.47 | 0.73 | 0.44 | 0.53 | 0.52 |
| **TOFU Forget05** | | | | | | | | | | | | | | |
| KL | 0.00 | 0.00 | 0.76 | 4.87e-12 | 0.01 | 0.00 | 0.16 | 0.00 | 0.26 | 0.26 | 0.00 | 0.27 | 0.26 | 0.00 |
| DPO | 0.35 | 0.85 | 0.49 | 3.17e-15 | 0.76 | 0.89 | 0.47 | 0.34 | 0.37 | 0.43 | 0.72 | 0.41 | 0.50 | 0.49 |
| NPO | 0.45 | 0.17 | 0.61 | 3.64e-05 | 0.46 | 0.19 | 0.38 | 0.37 | 0.33 | 0.40 | 0.68 | 0.36 | 0.43 | 0.36 |
| GA | 0.00 | 0.00 | 0.76 | 2.17e-13 | 0.01 | 0.00 | 0.16 | 0.00 | 0.26 | 0.26 | 0.00 | 0.26 | 0.25 | 0.00 |
| GA+FILA | 0.00 | 0.00 | 0.22 | 4.77e-17 | 0.00 | 0.00 | 0.35 | 0.00 | 0.22 | 0.35 | 0.00 | 0.19 | 0.37 | 0.00 |
| GD | 0.24 | 0.03 | 0.56 | 1.76e-03 | 0.32 | 0.18 | 0.44 | 0.06 | 0.35 | 0.41 | 0.39 | 0.35 | 0.43 | 0.23 |
| IHL | 0.45 | 0.42 | 0.50 | 4.18e-11 | 0.79 | 0.87 | 0.49 | 0.38 | 0.38 | 0.46 | 0.71 | 0.41 | 0.50 | 0.51 |
| GD+FILA | 0.04 | 0.00 | 0.71 | 4.16e-06 | 0.05 | 0.00 | 0.17 | 0.00 | 0.26 | 0.23 | 0.02 | 0.27 | 0.28 | 0.00 |
| IHL+FILA | 0.34 | 0.04 | 0.60 | 3.02e-03 | 0.71 | 0.81 | 0.48 | 0.37 | 0.38 | 0.46 | 0.69 | 0.42 | 0.52 | 0.50 |
| **TOFU Forget10** | | | | | | | | | | | | | | |
| KL | 0.01 | 0.00 | 0.60 | 2.17e-06 | 0.01 | 0.00 | 0.17 | 0.00 | 0.29 | 0.43 | 0.00 | 0.25 | 0.40 | 0.00 |
| DPO | 0.28 | 0.85 | 0.48 | 2.51e-18 | 0.81 | 0.91 | 0.48 | 0.32 | 0.37 | 0.43 | 0.71 | 0.41 | 0.49 | 0.49 |
| NPO | 0.44 | 0.25 | 0.65 | 2.31e-03 | 0.45 | 0.28 | 0.35 | 0.39 | 0.33 | 0.38 | 0.67 | 0.37 | 0.42 | 0.38 |
| GA | 0.01 | 0.00 | 0.60 | 2.17e-06 | 0.01 | 0.00 | 0.17 | 0.00 | 0.28 | 0.42 | 0.00 | 0.25 | 0.39 | 0.00 |
| GA+FILA | 0.00 | 0.00 | 0.23 | 4.22e-21 | 0.00 | 0.00 | 0.33 | 0.00 | 0.29 | 0.53 | 0.00 | 0.27 | 0.44 | 0.00 |
| GD | 0.11 | 0.00 | 0.45 | 3.33e-06 | 0.39 | 0.34 | 0.42 | 0.09 | 0.36 | 0.53 | 0.34 | 0.41 | 0.53 | 0.29 |
| IHL | 0.34 | 0.20 | 0.53 | 2.89e-11 | 0.81 | 0.86 | 0.50 | 0.42 | 0.39 | 0.47 | 0.70 | 0.42 | 0.53 | 0.52 |
| GD+FILA | 0.10 | 0.00 | 0.43 | 2.02e-08 | 0.10 | 0.00 | 0.27 | 0.00 | 0.25 | 0.38 | 0.03 | 0.27 | 0.40 | 0.00 |
| IHL+FILA | 0.13 | 0.02 | 0.68 | 2.08e-02 | 0.66 | 0.79 | 0.46 | 0.42 | 0.38 | 0.46 | 0.72 | 0.44 | 0.52 | 0.51 |

Table 5: Detailed TOFU Results using the Phi-1.5B model with LoRA rank = 32.

| Method | Forget Quality | | | | Model Utility | | | | | | | | | | |
|---|---|---|---|---|---|---|---|---|---|---|---|---|---|---|---|
| | Forget Set | | | FQ (↑) | Retain Set | | | Real Authors | | | Real World | | | MU (↑) |
| | Rouge-L | Prob. | Truth Ratio | | Rouge-L | Prob. | Truth Ratio | Rouge-L | Prob. | Truth Ratio | Rouge-L | Prob. | Truth Ratio | |
| Original | 0.99 | 0.99 | 0.51 | 2.19e-20 | 0.98 | 0.99 | 0.47 | 0.94 | 0.47 | 0.62 | 0.89 | 0.43 | 0.55 | 0.63 |
| Retain90 | 0.40 | 0.15 | 0.67 | 1.00e+00 | 0.98 | 0.99 | 0.47 | 0.92 | 0.47 | 0.61 | 0.88 | 0.42 | 0.55 | 0.63 |
| **TOFU Forget01** | | | | | | | | | | | | | | |
| KL | 0.95 | 0.99 | 0.55 | 9.73e-05 | 0.98 | 0.99 | 0.47 | 0.94 | 0.47 | 0.62 | 0.90 | 0.43 | 0.56 | 0.63 |
| DPO | 0.95 | 0.99 | 0.56 | 1.40e-04 | 0.98 | 0.99 | 0.47 | 0.93 | 0.47 | 0.62 | 0.89 | 0.43 | 0.55 | 0.63 |
| NPO | 0.95 | 0.99 | 0.55 | 9.73e-05 | 0.98 | 0.99 | 0.47 | 0.93 | 0.47 | 0.62 | 0.89 | 0.43 | 0.56 | 0.63 |
| GA | 0.95 | 0.99 | 0.55 | 6.71e-05 | 0.98 | 0.99 | 0.47 | 0.94 | 0.47 | 0.62 | 0.89 | 0.43 | 0.56 | 0.63 |
| GA+FILA | 0.03 | 0.00 | 0.87 | 3.12e-05 | 0.04 | 0.00 | 0.12 | 0.01 | 0.25 | 0.22 | 0.01 | 0.27 | 0.25 | 0.00 |
| GD | 0.95 | 0.99 | 0.55 | 1.40e-04 | 0.98 | 0.99 | 0.47 | 0.94 | 0.47 | 0.62 | 0.89 | 0.43 | 0.55 | 0.63 |
| IHL | 0.95 | 0.99 | 0.55 | 1.17e-04 | 0.98 | 0.99 | 0.47 | 0.94 | 0.47 | 0.62 | 0.90 | 0.43 | 0.55 | 0.63 |
| GD+FILA | 0.03 | 0.00 | 0.87 | 1.15e-05 | 0.04 | 0.00 | 0.13 | 0.00 | 0.25 | 0.21 | 0.02 | 0.27 | 0.24 | 0.00 |
| IHL+FILA | 0.69 | 0.84 | 0.55 | 1.53e-04 | 0.98 | 0.99 | 0.47 | 0.93 | 0.46 | 0.60 | 0.89 | 0.42 | 0.54 | 0.62 |
| **TOFU Forget05** | | | | | | | | | | | | | | |
| KL | 0.92 | 0.96 | 0.53 | 9.25e-17 | 0.97 | 0.98 | 0.46 | 0.93 | 0.48 | 0.63 | 0.90 | 0.45 | 0.57 | 0.64 |
| DPO | 0.83 | 0.95 | 0.57 | 8.99e-14 | 0.86 | 0.95 | 0.44 | 0.92 | 0.47 | 0.60 | 0.87 | 0.45 | 0.56 | 0.62 |
| NPO | 0.89 | 0.95 | 0.54 | 2.47e-16 | 0.95 | 0.97 | 0.46 | 0.94 | 0.48 | 0.63 | 0.90 | 0.45 | 0.57 | 0.64 |
| GA | 0.90 | 0.96 | 0.54 | 6.50e-16 | 0.96 | 0.98 | 0.46 | 0.94 | 0.48 | 0.63 | 0.90 | 0.45 | 0.57 | 0.64 |
| GA+FILA | 0.01 | 0.00 | 0.83 | 1.23e-15 | 0.01 | 0.00 | 0.10 | 0.00 | 0.24 | 0.17 | 0.00 | 0.26 | 0.24 | 0.00 |
| GD | 0.93 | 0.97 | 0.52 | 6.50e-16 | 0.98 | 0.99 | 0.47 | 0.94 | 0.48 | 0.62 | 0.89 | 0.44 | 0.56 | 0.64 |
| IHL | 0.94 | 0.97 | 0.52 | 6.64e-17 | 0.98 | 0.99 | 0.47 | 0.94 | 0.47 | 0.62 | 0.90 | 0.44 | 0.56 | 0.64 |
| GD+FILA | 0.02 | 0.00 | 0.77 | 1.50e-08 | 0.03 | 0.00 | 0.14 | 0.01 | 0.24 | 0.17 | 0.00 | 0.24 | 0.21 | 0.00 |
| IHL+FILA | 0.54 | 0.70 | 0.58 | 6.87e-13 | 0.90 | 0.94 | 0.45 | 0.92 | 0.48 | 0.62 | 0.89 | 0.47 | 0.60 | 0.64 |
| **TOFU Forget10** | | | | | | | | | | | | | | |
| KL | 0.47 | 0.32 | 0.65 | 2.56e-05 | 0.47 | 0.33 | 0.35 | 0.93 | 0.43 | 0.55 | 0.89 | 0.43 | 0.56 | 0.49 |
| DPO | 0.45 | 0.94 | 0.55 | 5.10e-17 | 0.66 | 0.96 | 0.44 | 0.82 | 0.43 | 0.54 | 0.87 | 0.42 | 0.51 | 0.57 |
| NPO | 0.54 | 0.31 | 0.65 | 3.33e-06 | 0.54 | 0.31 | 0.35 | 0.94 | 0.39 | 0.50 | 0.90 | 0.39 | 0.51 | 0.47 |
| GA | 0.49 | 0.15 | 0.66 | 2.31e-03 | 0.49 | 0.15 | 0.33 | 0.93 | 0.39 | 0.51 | 0.91 | 0.37 | 0.50 | 0.39 |
| GA+FILA | 0.02 | 0.00 | 0.86 | 5.40e-18 | 0.02 | 0.00 | 0.09 | 0.00 | 0.24 | 0.19 | 0.00 | 0.24 | 0.18 | 0.00 |
| GD | 0.82 | 0.92 | 0.51 | 2.19e-16 | 0.92 | 0.97 | 0.47 | 0.92 | 0.46 | 0.60 | 0.88 | 0.42 | 0.55 | 0.62 |
| IHL | 0.73 | 0.87 | 0.57 | 3.71e-15 | 0.88 | 0.93 | 0.45 | 0.94 | 0.49 | 0.64 | 0.89 | 0.47 | 0.59 | 0.64 |
| GD+FILA | 0.01 | 0.00 | 0.85 | 1.83e-21 | 0.01 | 0.00 | 0.09 | 0.00 | 0.25 | 0.18 | 0.00 | 0.23 | 0.18 | 0.00 |
| IHL+FILA | 0.30 | 0.28 | 0.65 | 2.93e-01 | 0.91 | 0.96 | 0.45 | 0.89 | 0.47 | 0.62 | 0.88 | 0.45 | 0.57 | 0.63 |

Table 6: Detailed TOFU Results using the Llama2-7B model with LoRA rank = 4.

| Method | Forget Quality | | | | Model Utility | | | | | | | | | | |
|---|---|---|---|---|---|---|---|---|---|---|---|---|---|---|---|
| | Forget Set | | | FQ (↑) | Retain Set | | | Real Authors | | | Real World | | | MU (↑) |
| | Rouge-L | Prob. | Truth Ratio | | Rouge-L | Prob. | Truth Ratio | Rouge-L | Prob. | Truth Ratio | Rouge-L | Prob. | Truth Ratio | |
| Original | 0.99 | 0.99 | 0.51 | 2.19e-20 | 0.98 | 0.99 | 0.47 | 0.94 | 0.47 | 0.62 | 0.89 | 0.43 | 0.55 | 0.63 |
| Retain90 | 0.40 | 0.15 | 0.67 | 1.00e+00 | 0.98 | 0.99 | 0.47 | 0.92 | 0.47 | 0.61 | 0.88 | 0.42 | 0.55 | 0.63 |
| **TOFU Forget01** | | | | | | | | | | | | | | |
| KL | 0.95 | 0.98 | 0.56 | 1.17e-04 | 0.98 | 0.99 | 0.47 | 0.93 | 0.47 | 0.62 | 0.90 | 0.43 | 0.56 | 0.64 |
| DPO | 0.94 | 0.99 | 0.56 | 1.40e-04 | 0.98 | 0.99 | 0.47 | 0.94 | 0.47 | 0.62 | 0.90 | 0.43 | 0.56 | 0.63 |
| NPO | 0.94 | 0.98 | 0.55 | 6.71e-05 | 0.98 | 0.99 | 0.47 | 0.94 | 0.47 | 0.62 | 0.89 | 0.43 | 0.56 | 0.63 |
| GA | 0.94 | 0.98 | 0.55 | 6.71e-05 | 0.98 | 0.99 | 0.47 | 0.93 | 0.47 | 0.62 | 0.89 | 0.43 | 0.55 | 0.63 |
| GA+FILA | 0.01 | 0.00 | 0.86 | 3.12e-05 | 0.02 | 0.00 | 0.09 | 0.01 | 0.25 | 0.22 | 0.00 | 0.24 | 0.17 | 0.00 |
| GD | 0.95 | 0.99 | 0.55 | 1.17e-04 | 0.98 | 0.99 | 0.47 | 0.93 | 0.47 | 0.62 | 0.89 | 0.43 | 0.56 | 0.63 |
| IHL | 0.95 | 0.99 | 0.55 | 1.40e-04 | 0.98 | 0.99 | 0.47 | 0.94 | 0.47 | 0.61 | 0.90 | 0.43 | 0.55 | 0.63 |
| GD+FILA | 0.02 | 0.00 | 0.85 | 2.35e-06 | 0.02 | 0.00 | 0.09 | 0.00 | 0.25 | 0.22 | 0.00 | 0.24 | 0.19 | 0.00 |
| IHL+FILA | 0.57 | 0.68 | 0.55 | 2.60e-04 | 0.97 | 0.98 | 0.47 | 0.92 | 0.46 | 0.60 | 0.89 | 0.42 | 0.54 | 0.62 |
| **TOFU Forget05** | | | | | | | | | | | | | | |
| KL | 0.77 | 0.89 | 0.54 | 6.50e-16 | 0.88 | 0.95 | 0.45 | 0.94 | 0.47 | 0.61 | 0.91 | 0.45 | 0.57 | 0.63 |
| DPO | 0.14 | 0.90 | 0.58 | 6.87e-13 | 0.27 | 0.91 | 0.43 | 0.59 | 0.44 | 0.57 | 0.84 | 0.44 | 0.54 | 0.50 |
| NPO | 0.62 | 0.72 | 0.57 | 1.09e-14 | 0.68 | 0.76 | 0.44 | 0.87 | 0.46 | 0.61 | 0.88 | 0.46 | 0.57 | 0.60 |
| GA | 0.64 | 0.78 | 0.56 | 1.21e-13 | 0.71 | 0.82 | 0.44 | 0.89 | 0.47 | 0.62 | 0.90 | 0.46 | 0.57 | 0.61 |
| GA+FILA | 0.01 | 0.00 | 0.88 | 1.32e-19 | 0.01 | 0.00 | 0.07 | 0.01 | 0.25 | 0.19 | 0.00 | 0.24 | 0.17 | 0.00 |
| GD | 0.90 | 0.94 | 0.52 | 3.17e-15 | 0.97 | 0.98 | 0.47 | 0.92 | 0.48 | 0.62 | 0.89 | 0.43 | 0.56 | 0.64 |
| IHL | 0.84 | 0.94 | 0.53 | 1.24e-17 | 0.98 | 0.98 | 0.47 | 0.94 | 0.47 | 0.61 | 0.90 | 0.44 | 0.56 | 0.63 |
| GD+FILA | 0.02 | 0.00 | 0.86 | 4.46e-18 | 0.03 | 0.00 | 0.09 | 0.01 | 0.25 | 0.19 | 0.01 | 0.24 | 0.19 | 0.00 |
| IHL+FILA | 0.40 | 0.39 | 0.62 | 4.77e-07 | 0.86 | 0.94 | 0.45 | 0.92 | 0.48 | 0.63 | 0.91 | 0.45 | 0.59 | 0.63 |
| **TOFU Forget10** | | | | | | | | | | | | | | |
| KL | 0.31 | 0.00 | 0.65 | 7.31e-03 | 0.31 | 0.00 | 0.22 | 0.07 | 0.29 | 0.40 | 0.62 | 0.34 | 0.48 | 0.01 |
| DPO | 0.34 | 0.94 | 0.54 | 5.10e-17 | 0.67 | 0.96 | 0.45 | 0.65 | 0.42 | 0.53 | 0.84 | 0.40 | 0.49 | 0.55 |
| NPO | 0.50 | 0.26 | 0.67 | 1.73e-05 | 0.51 | 0.26 | 0.33 | 0.91 | 0.39 | 0.51 | 0.91 | 0.38 | 0.49 | 0.45 |
| GA | 0.11 | 0.00 | 0.65 | 3.32e-08 | 0.12 | 0.00 | 0.17 | 0.00 | 0.27 | 0.38 | 0.10 | 0.28 | 0.42 | 0.00 |
| GA+FILA | 0.00 | 0.00 | 0.61 | 3.32e-08 | 0.00 | 0.00 | 0.15 | 0.00 | 0.26 | 0.36 | 0.00 | 0.21 | 0.24 | 0.00 |
| GD | 0.47 | 0.47 | 0.49 | 3.71e-15 | 0.52 | 0.67 | 0.48 | 0.87 | 0.44 | 0.58 | 0.87 | 0.40 | 0.56 | 0.56 |
| IHL | 0.59 | 0.77 | 0.58 | 2.86e-14 | 0.86 | 0.91 | 0.45 | 0.93 | 0.51 | 0.66 | 0.89 | 0.48 | 0.62 | 0.65 |
| GD+FILA | 0.03 | 0.00 | 0.85 | 1.15e-17 | 0.04 | 0.00 | 0.07 | 0.01 | 0.25 | 0.21 | 0.01 | 0.23 | 0.16 | 0.00 |
| IHL+FILA | 0.15 | 0.11 | 0.74 | 3.63e-07 | 0.93 | 0.97 | 0.44 | 0.86 | 0.49 | 0.64 | 0.86 | 0.45 | 0.59 | 0.63 |

Table 7: Detailed TOFU Results using the Llama2-7B model with LoRA rank = 8.

| Method | Forget Quality | | | | Model Utility | | | | | | | | | | |
| --- | --- | --- | --- | --- | --- | --- | --- | --- | --- | --- | --- | --- | --- | --- |
| | Forget Set | | | FQ (↑) | Retain Set | | | Real Authors | | | Real World | | | MU (↑) |
| | Rouge-L | Prob. | Truth Ratio | | Rouge-L | Prob. | Truth Ratio | Rouge-L | Prob. | Truth Ratio | Rouge-L | Prob. | Truth Ratio | |
| Original | 0.99 | 0.99 | 0.51 | 2.19e-20 | 0.98 | 0.99 | 0.47 | 0.94 | 0.47 | 0.62 | 0.89 | 0.43 | 0.55 | 0.63 |
| Retain90 | 0.40 | 0.15 | 0.67 | 1.00e+00 | 0.98 | 0.99 | 0.47 | 0.92 | 0.47 | 0.61 | 0.88 | 0.42 | 0.55 | 0.63 |
| **TOFU Forget01** | | | | | | | | | | | | | | |
| KL | 0.90 | 0.96 | 0.56 | 1.07e-04 | 0.98 | 0.99 | 0.47 | 0.93 | 0.48 | 0.62 | 0.89 | 0.44 | 0.56 | 0.64 |
| DPO | 0.94 | 0.99 | 0.56 | 3.37e-04 | 0.97 | 0.99 | 0.46 | 0.94 | 0.48 | 0.62 | 0.90 | 0.44 | 0.55 | 0.63 |
| NPO | 0.89 | 0.96 | 0.55 | 8.09e-05 | 0.98 | 0.99 | 0.47 | 0.93 | 0.47 | 0.62 | 0.89 | 0.43 | 0.56 | 0.64 |
| GA | 0.90 | 0.96 | 0.55 | 9.73e-05 | 0.98 | 0.99 | 0.47 | 0.93 | 0.48 | 0.62 | 0.90 | 0.43 | 0.56 | 0.64 |
| GA+FILA | 0.00 | 0.00 | 0.69 | 1.58e-02 | 0.01 | 0.00 | 0.17 | 0.00 | 0.27 | 0.24 | 0.01 | 0.26 | 0.21 | 0.00 |
| GD | 0.92 | 0.97 | 0.55 | 1.28e-04 | 0.98 | 0.99 | 0.47 | 0.94 | 0.47 | 0.62 | 0.89 | 0.43 | 0.56 | 0.63 |
| IHL | 0.88 | 0.97 | 0.55 | 1.40e-04 | 0.98 | 0.99 | 0.47 | 0.93 | 0.47 | 0.61 | 0.90 | 0.43 | 0.55 | 0.63 |
| GD+FILA | 0.02 | 0.00 | 0.67 | 3.33e-01 | 0.01 | 0.00 | 0.18 | 0.00 | 0.27 | 0.27 | 0.01 | 0.28 | 0.25 | 0.00 |
| IHL+FILA | 0.52 | 0.42 | 0.56 | 7.13e-04 | 0.91 | 0.97 | 0.47 | 0.89 | 0.46 | 0.61 | 0.88 | 0.42 | 0.55 | 0.62 |
| **TOFU Forget05** | | | | | | | | | | | | | | |
| KL | 0.51 | 0.52 | 0.58 | 2.80e-12 | 0.54 | 0.58 | 0.43 | 0.90 | 0.41 | 0.52 | 0.91 | 0.42 | 0.53 | 0.54 |
| DPO | 0.17 | 0.91 | 0.57 | 1.62e-13 | 0.44 | 0.93 | 0.44 | 0.60 | 0.43 | 0.55 | 0.84 | 0.42 | 0.51 | 0.53 |
| NPO | 0.47 | 0.48 | 0.63 | 8.80e-10 | 0.48 | 0.52 | 0.39 | 0.89 | 0.44 | 0.57 | 0.90 | 0.45 | 0.57 | 0.53 |
| GA | 0.48 | 0.45 | 0.62 | 1.12e-09 | 0.50 | 0.50 | 0.39 | 0.90 | 0.41 | 0.52 | 0.90 | 0.43 | 0.54 | 0.52 |
| GA+FILA | 0.03 | 0.00 | 0.68 | 1.00e-03 | 0.03 | 0.00 | 0.15 | 0.01 | 0.24 | 0.22 | 0.02 | 0.22 | 0.19 | 0.00 |
| GD | 0.68 | 0.79 | 0.51 | 4.96e-14 | 0.82 | 0.94 | 0.47 | 0.90 | 0.46 | 0.59 | 0.86 | 0.41 | 0.56 | 0.61 |
| IHL | 0.69 | 0.84 | 0.54 | 3.42e-16 | 0.94 | 0.97 | 0.46 | 0.94 | 0.46 | 0.59 | 0.90 | 0.44 | 0.56 | 0.63 |
| GD+FILA | 0.05 | 0.00 | 0.80 | 4.19e-10 | 0.06 | 0.00 | 0.12 | 0.01 | 0.26 | 0.24 | 0.02 | 0.26 | 0.22 | 0.00 |
| IHL+FILA | 0.25 | 0.18 | 0.69 | 1.34e-02 | 0.88 | 0.95 | 0.46 | 0.93 | 0.47 | 0.62 | 0.89 | 0.44 | 0.58 | 0.63 |
| **TOFU Forget10** | | | | | | | | | | | | | | |
| KL | 0.00 | 0.00 | 0.81 | 2.86e-14 | 0.00 | 0.00 | 0.09 | 0.00 | 0.22 | 0.23 | 0.00 | 0.23 | 0.22 | 0.00 |
| DPO | 0.27 | 0.94 | 0.53 | 1.06e-16 | 0.73 | 0.97 | 0.45 | 0.58 | 0.40 | 0.52 | 0.85 | 0.39 | 0.48 | 0.54 |
| NPO | 0.47 | 0.26 | 0.70 | 1.66e-04 | 0.47 | 0.27 | 0.31 | 0.91 | 0.41 | 0.53 | 0.90 | 0.38 | 0.50 | 0.44 |
| GA | 0.00 | 0.00 | 0.79 | 1.60e-11 | 0.00 | 0.00 | 0.11 | 0.00 | 0.22 | 0.23 | 0.00 | 0.22 | 0.24 | 0.00 |
| GA+FILA | 0.00 | 0.00 | 0.90 | 4.07e-32 | 0.00 | 0.00 | 0.04 | 0.00 | 0.25 | 0.17 | 0.00 | 0.24 | 0.13 | 0.00 |
| GD | 0.39 | 0.06 | 0.47 | 7.41e-13 | 0.44 | 0.19 | 0.52 | 0.81 | 0.42 | 0.58 | 0.86 | 0.38 | 0.55 | 0.44 |
| IHL | 0.43 | 0.52 | 0.63 | 5.73e-07 | 0.89 | 0.94 | 0.45 | 0.90 | 0.52 | 0.67 | 0.89 | 0.48 | 0.62 | 0.65 |
| GD+FILA | 0.04 | 0.00 | 0.70 | 2.65e-02 | 0.04 | 0.00 | 0.16 | 0.00 | 0.27 | 0.31 | 0.02 | 0.23 | 0.27 | 0.00 |
| IHL+FILA | 0.04 | 0.02 | 0.79 | 8.69e-10 | 0.91 | 0.97 | 0.44 | 0.88 | 0.50 | 0.65 | 0.87 | 0.46 | 0.59 | 0.64 |

Table 8: Detailed TOFU Results using the Llama2-7B model with LoRA rank = 16.

| Method | Forget Quality | | | | Model Utility | | | | | | | | | | |
| --- | --- | --- | --- | --- | --- | --- | --- | --- | --- | --- | --- | --- | --- | --- |
| | Forget Set | | | FQ (↑) | Retain Set | | | Real Authors | | | Real World | | | MU (↑) |
| | Rouge-L | Prob. | Truth Ratio | | Rouge-L | Prob. | Truth Ratio | Rouge-L | Prob. | Truth Ratio | Rouge-L | Prob. | Truth Ratio | |
| Original | 0.99 | 0.99 | 0.51 | 2.19e-20 | 0.98 | 0.99 | 0.47 | 0.94 | 0.47 | 0.62 | 0.89 | 0.43 | 0.55 | 0.63 |
| Retain90 | 0.40 | 0.15 | 0.67 | 1.00e+00 | 0.98 | 0.99 | 0.47 | 0.92 | 0.47 | 0.61 | 0.88 | 0.42 | 0.55 | 0.63 |
| **TOFU Forget01** | | | | | | | | | | | | | | |
| KL | 0.78 | 0.90 | 0.56 | 5.05e-05 | 0.96 | 0.98 | 0.47 | 0.93 | 0.48 | 0.62 | 0.90 | 0.44 | 0.56 | 0.64 |
| DPO | 0.88 | 0.97 | 0.58 | 6.57e-04 | 0.90 | 0.97 | 0.46 | 0.93 | 0.48 | 0.63 | 0.90 | 0.45 | 0.56 | 0.63 |
| NPO | 0.73 | 0.90 | 0.55 | 1.40e-04 | 0.95 | 0.98 | 0.47 | 0.93 | 0.48 | 0.63 | 0.90 | 0.44 | 0.56 | 0.64 |
| GA | 0.76 | 0.90 | 0.56 | 8.87e-05 | 0.96 | 0.98 | 0.47 | 0.93 | 0.47 | 0.62 | 0.90 | 0.44 | 0.56 | 0.63 |
| GA+FILA | 0.03 | 0.00 | 0.77 | 7.70e-03 | 0.02 | 0.00 | 0.13 | 0.00 | 0.25 | 0.21 | 0.00 | 0.23 | 0.18 | 0.00 |
| GD | 0.82 | 0.91 | 0.55 | 1.53e-04 | 0.96 | 0.98 | 0.47 | 0.93 | 0.47 | 0.62 | 0.88 | 0.43 | 0.56 | 0.63 |
| IHL | 0.76 | 0.89 | 0.55 | 2.18e-04 | 0.96 | 0.98 | 0.47 | 0.93 | 0.47 | 0.61 | 0.90 | 0.43 | 0.55 | 0.63 |
| GD+FILA | 0.04 | 0.00 | 0.81 | 1.67e-04 | 0.03 | 0.00 | 0.14 | 0.00 | 0.24 | 0.21 | 0.01 | 0.25 | 0.19 | 0.00 |
| IHL+FILA | 0.43 | 0.20 | 0.61 | 4.13e-03 | 0.81 | 0.94 | 0.47 | 0.89 | 0.45 | 0.60 | 0.88 | 0.42 | 0.55 | 0.61 |
| **TOFU Forget05** | | | | | | | | | | | | | | |
| KL | 0.45 | 0.09 | 0.64 | 4.81e-04 | 0.46 | 0.12 | 0.36 | 0.91 | 0.34 | 0.45 | 0.90 | 0.36 | 0.50 | 0.36 |
| DPO | 0.30 | 0.93 | 0.56 | 2.72e-14 | 0.77 | 0.96 | 0.44 | 0.78 | 0.42 | 0.54 | 0.86 | 0.41 | 0.50 | 0.57 |
| NPO | 0.46 | 0.29 | 0.70 | 1.81e-05 | 0.47 | 0.33 | 0.32 | 0.94 | 0.40 | 0.51 | 0.91 | 0.43 | 0.54 | 0.47 |
| GA | 0.44 | 0.09 | 0.69 | 1.14e-01 | 0.44 | 0.11 | 0.29 | 0.85 | 0.37 | 0.49 | 0.92 | 0.37 | 0.50 | 0.34 |
| GA+FILA | 0.01 | 0.00 | 0.86 | 2.47e-16 | 0.02 | 0.00 | 0.09 | 0.00 | 0.24 | 0.20 | 0.01 | 0.24 | 0.17 | 0.00 |
| GD | 0.40 | 0.30 | 0.50 | 6.87e-13 | 0.49 | 0.62 | 0.47 | 0.82 | 0.41 | 0.53 | 0.86 | 0.41 | 0.56 | 0.54 |
| IHL | 0.51 | 0.63 | 0.56 | 2.12e-12 | 0.88 | 0.94 | 0.46 | 0.93 | 0.47 | 0.60 | 0.90 | 0.44 | 0.57 | 0.63 |
| GD+FILA | 0.11 | 0.00 | 0.68 | 1.00e-03 | 0.11 | 0.00 | 0.18 | 0.01 | 0.24 | 0.27 | 0.03 | 0.21 | 0.23 | 0.00 |
| IHL+FILA | 0.20 | 0.05 | 0.75 | 7.38e-03 | 0.87 | 0.95 | 0.48 | 0.91 | 0.45 | 0.59 | 0.88 | 0.44 | 0.58 | 0.62 |
| **TOFU Forget10** | | | | | | | | | | | | | | |
| KL | 0.00 | 0.00 | 0.86 | 4.22e-21 | 0.00 | 0.00 | 0.07 | 0.00 | 0.26 | 0.25 | 0.00 | 0.25 | 0.23 | 0.00 |
| DPO | 0.23 | 0.94 | 0.52 | 5.10e-17 | 0.88 | 0.98 | 0.46 | 0.75 | 0.40 | 0.50 | 0.86 | 0.38 | 0.47 | 0.56 |
| NPO | 0.47 | 0.28 | 0.71 | 4.69e-04 | 0.48 | 0.31 | 0.31 | 0.87 | 0.43 | 0.56 | 0.89 | 0.40 | 0.52 | 0.46 |
| GA | 0.00 | 0.00 | 0.60 | 5.07e-06 | 0.00 | 0.00 | 0.20 | 0.00 | 0.19 | 0.28 | 0.00 | 0.20 | 0.35 | 0.00 |
| GA+FILA | 0.00 | 0.00 | 0.67 | 3.33e-06 | 0.00 | 0.00 | 0.18 | 0.00 | 0.26 | 0.32 | 0.00 | 0.31 | 0.39 | 0.00 |
| GD | 0.01 | 0.00 | 0.56 | 5.44e-08 | 0.18 | 0.20 | 0.65 | 0.43 | 0.48 | 0.63 | 0.81 | 0.43 | 0.58 | 0.38 |
| IHL | 0.14 | 0.11 | 0.79 | 1.22e-08 | 0.94 | 0.97 | 0.45 | 0.88 | 0.48 | 0.62 | 0.88 | 0.45 | 0.58 | 0.63 |
| GD+FILA | 0.03 | 0.00 | 0.83 | 9.16e-16 | 0.04 | 0.00 | 0.11 | 0.00 | 0.24 | 0.20 | 0.01 | 0.25 | 0.21 | 0.00 |
| IHL+FILA | 0.04 | 0.02 | 0.73 | 3.77e-05 | 0.90 | 0.96 | 0.48 | 0.92 | 0.52 | 0.67 | 0.87 | 0.46 | 0.60 | 0.65 |

Table 9: Detailed TOFU Results using the Llama2-7B model with LoRA rank = 32.

