# OpenReview forum: "Towards Robust and Parameter-Efficient Knowledge Unlearning for LLMs"
_ICLR.cc/2025/Conference — ICLR 2025 Poster_

### Official Review · Reviewer_AAXs · 2024-10-21

**Soundness:** 3
**Presentation:** 3
**Contribution:** 3
**Rating:** 6
**Confidence:** 4

**Summary:**

The paper focus on the problem of unstable optimization, catastrophic forgetting, and computational cost from Gradient Ascent for LLM unlearning, and propose two novel techniques, including the Inverted Hinge Loss and Fisher Information weighted initialization of LoRA adapters, for robust and efficient unlearning for LLMs. Experiments on two tasks with different LLMs show that the proposed methods enable faster and more stable LoRA-based LLM unlearning.

**Strengths:**

1. The paper makes a good contribution to knowledge unlearning of LLMs through improving tuning stability and efficiency with Inverted Hinge Loss and Fisher-Initialization of Low-rank Adapters, respectively. The proposed method is valuable of managing unnecessary forgetting and unbounded optimization in typical Gradient Ascent strategies.
2. The paper is generally well-written, particularly in formula derivation and clear explanation about IHL and FILA solve the weaknesses of GA in Section 3.3 and Section 3.4.
3. The experiments and analysis of the paper is comprehensive, with great illustration on performance evaluation using high quality charts.

**Weaknesses:**

1. In Introduction Section (Line 51-53), you mention GA and its shortcomings, I think a better way of writing here would be providing a brief overview of 2-3 other key knowledge unlearning approaches beyond GA, and summarize 1-2 common shortcomings across these methods that motivate the proposed approach. GA should be only one of those existing typical methods.

2. In Introduction Section (Line 76), you mention the application of LoRA to LLM unlearning remains unexplored, however, there are some existing studies using LoRA for LLM unlearning, including Practical Unlearning for Large Language Models (https://arxiv.org/abs/2407.10223) and Machine Unlearning in Large Language Models (https://arxiv.org/abs/2405.15152v1). It would be better to briefly summarize (in 1-2 sentences each) how LoRA was used for unlearning in these two papers, and then explain how their proposed approach differs or improves upon these methods.

3. Some important content is missing. In Introduction Section, a lack of clear summarization of contributions in the paper, making readers difficult to capture the important points of the study. Besides, in Related Work Section, a brief comparison between your proposed method and other relevant studies should be presented to better emphasize the advantages of your work.

4. In Section 3.3 (Line 223-227), you should provide a more detailed explanation of how you arrived at these hypotheses. There is still a gap between GA motivation and its weaknesses. To make the illustration more convincible here, a better way would be providing a specific example or mathematical derivation showing how the GA loss function leads to one of the stated problems (e.g., unbounded optimization or inefficient gradient updates).

5. In Section 3.4 (Line 265), a basic and brief description of Fisher Information is necessary here for better understanding of the reason you employ it to address the importance quantification you mentioned before.

6. In Table 1, to make the table more readable, the data of important results should be highlighted via bolding the best result in each row or using color coding to show relative performance between methods, in order to show either your FILA LoRA performs much better than traditional LoRA, or it can approach the performance of full fine-tuning.

7. There are some writing flaws:
* Some sentences are too long to follow and comprehend, such as Line 89-93.
* In Line 419-421, there are two "Second" in these two sentences, making them difficult to be understood.
* The capitalization in the text should be more consistent. For instance, you use lowercase "l" in "Inverted Hinge loss" at Line 20 and Line 161, but uppercase "L" in "Inverted Hinge Loss" at Line 82. All uppercase would be better.

**Questions:**

1. In Introduction Section (Line 72-74), can you explain more about the reason that low-rankness can be beneficial in stabilizing optimization and preventing catastrophic forgetting? Clearer illustration would be better here.

2. In Table 1, why you record results from running different epochs? Does it mean the method reaches the optimal with these epochs?

3. In Experiments Section, why different LLMs are used for those two tasks? Have you evaluated more popular and larger LLMs such as Llama3.1-8B? I suggest giving explanation of the strategy and purpose of model selection.

---

> ### Author Response · Authors · 2024-11-19
>
> > [W1] In Introduction Section (Line 51-53), you mention GA and its shortcomings, I think a better way of writing here would be providing a brief overview of 2-3 other key knowledge unlearning approaches beyond GA, and summarize 1-2 common shortcomings across these methods that motivate the proposed approach. GA should be only one of those existing typical methods.
>
> &rarr; We thank the reviewer for the suggestion. We will revise the introduction section to share other representative approaches such as KL- and preference optimization-based methods for unlearning, then proceed to motivate our approaches.
>
> ---
> > [W2] In Introduction Section (Line 76), you mention the application of LoRA to LLM unlearning remains unexplored, however, there are some existing studies using LoRA for LLM unlearning. It would be better to briefly summarize how LoRA was used for unlearning in these two papers, and then explain how their proposed approach differs or improves upon these methods.
>
> &rarr; Thank you for sharing additional literature. Upon review, we have found that the first paper proposes orthogonal low-rank adaptation tailored for continual unlearning scenarios [A]. While using LoRA in the experiments, the second paper contributes mainly on testing a new datasets and evaluation metric [B]. In contrast, our work specifically focuses on the stability vs. plasticity trade-off within LoRA-based unlearning, and proposes a novel unlearning loss function and LoRA-initialization towards optimizing this trade-off. We will revise the introduction and related work sections to refer suggested papers.
>
> ---
> > [W3] In Introduction Section, a lack of clear summarization of contributions in the paper, making readers difficult to capture the important points of the study. In Related Work Section, a brief comparison between your proposed method and other relevant studies should be presented to better emphasize the advantages of your work.
>
> &rarr; We will revise the related work section to add comparisons of previous work vs. our proposed methods.
>
> ---
> > [W4] In Section 3.3 (Line 223-227), you should provide a more detailed explanation of how you arrived at these hypotheses. There is still a gap between GA motivation and its weaknesses. To make the illustration more convincible here, a better way would be providing a specific example or mathematical derivation showing how the GA loss function leads to one of the stated problems (e.g., unbounded optimization or inefficient gradient updates).
>
> &rarr; Thank you for your insightful comments. Here we provide additional clarification regarding the derivative (Line 218) and its relation to the three issues of GA.
>
> Given the prefix $x\_{<t}$, GA reduces the prediction score of the true token $x_t$ by an amount proportional to $1 - p_\theta(x_t | x_{<t})$ and increases the scores of other tokens $v \neq x_t$ by $p_\theta(v | x_{<t})$. This process effectively shifts the prediction given $x_{<t}$ away from the true token $x_t$, achieving unlearning. Based on this analysis, we share three key issues of GA:
> 1. **Gradient Spread:** GA reduces the true token's score while increasing scores of all other tokens. When using large vocabulary sizes as in most LLMs, this leads to gradients primarily boosting tokens other than the true token $x_t$.
> 2. **Unbounded Loss:** With LLMs, GA reduces $\log(p_\theta(x_t | x_{<t}))$ by maximizing the cross-entropy loss, which by nature of entropy, is an unbounded optimization problem. This implies the possibility of divergence as unlearning progresses.
> 3. **Performance Degradation:** Each sequence $x$ in the forget set $\mathcal{D}\_f$ can require different number of model updates for successful unlearning, yet GA applies the same gradient updates to decrease $p\_\theta(x_t | x_{<t})$ at every iteration regardless of this distinction. This leads to unnecessary model updates, which we found to induce catastrophic forgetting of knowledge and generative performance.
>
> Our novel objective function IHL is specifically designed to mitigate these limitations (Lines 244-252). We will further clarify this discussion in the revised manuscript.
>
> ---
> > [W5] In Section 3.4 (Line 265), a basic and brief description of Fisher Information is necessary here for better understanding of the reason you employ it to address the importance quantification you mentioned before.
>
> &rarr; Mathematically, the Fisher Information equals the variance of the partial derivative of the log likelihood with respect to the model parameter (shown in Equation 2). Intuitively, the Fisher Information can be considered a measurement on *how much the LLM output changes following a small change on the model weights*. We will revise the draft for better understanding of our choice in measuring weight importances.

---

> > ### Author Response · Authors · 2024-11-19
> >
> > > [W6] In Table 1, to make the table more readable, the data of important results should be highlighted via bolding the best result in each row or using color coding to show relative performance between methods.
> >
> > &rarr; We agree with the reviewer. We will revise Table 1 to highlight best results per model size and also add relative performance gains/losses with color coding for better readability.
> >
> > ---
> > > [W7] There are some writing flaws.
> >
> > &rarr; We thank the reviewer for spotting these writing errors. We will revise the draft to correct the errors.
> >
> > ---
> > > [Q1] In Introduction Section (Line 72-74), can you explain more about the reason that low-rankness can be beneficial in stabilizing optimization and preventing catastrophic forgetting? Clearer illustration would be better here.
> >
> > &rarr; Intuitively, the weight changes that can be made within the LLM is constrained by the low-rank structure of LoRA. Compared to full parameter fine-tuning, this constraint effectively prevents the model from "changing too much" from the pretrained model, thereby better retaining previously acquired knowledge. As we will revise Figure 1 to illustrate our methods, we will also aim to depict this notion of stability from low-rankness in the figure.
> >
> > ---
> > > [Q2] In Table 1, why you record results from running different epochs? Does it mean the method reaches the optimal with these epochs?
> >
> > &rarr; For the TDEC experiments shown in Table 1, we follow previous work [A] and stop the unlearning process when the model meets a threshold based on $n$-gram and token-wise overlaps against the unlearning target sequences: after each unlearning epoch, we measure (1) the $n$-gram Extraction Likelihood and (2) Memorization Accuracy of the model on unlearning target sequences, then compare those to the same measurements obtained from a held-out validation set. If the values are smaller than those from the held-out set, it means the model generates target sequences similarly to unforeseen sequences, indicating successful unlearning.
> >
> > ---
> > > [Q3] In Experiments Section, why different LLMs are used for those two tasks? Have you evaluated more popular and larger LLMs such as Llama3.1-8B? I suggest giving explanation of the strategy and purpose of model selection.
> >
> > &rarr; For our TDEC experiments, we choose to use the GPT-Neo family following previous work [C]. Note that because TDEC consists of unlearning targets chosen from the Pile dataset [D], our model choice is limited to those known to be pretrained on the Pile corpus, which excludes more recent larger LLMs such as Llama3.1-8B.
> >
> > For our TOFU experiments, we use Phi-1.5B and Llama2-7B models following the original benchmark paper [E]. Not to mention the easy reproducibility from publicly available base models from which we run unlearning, this also ensures that our empirical findings are directly comparable with original results from the TOFU paper [E] as well as results from papers that experiment unlearning on TOFU [A, F].
> >
> > [A] [Gao et al., Practical Unlearning for Large Language Models. arXiv 2024.](https://arxiv.org/abs/2407.10223)
> >
> > [B] [Gundavarapu et al., Machine Unlearning in Large Language Models. arXiv 2024.](https://arxiv.org/abs/2405.15152)
> >
> > [C] [Jang et al., Knowledge unlearning for mitigating privacy risks in language models. ACL 2023.](https://arxiv.org/abs/2210.01504)
> >
> > [D] [Gao et al., The Pile: An 800GB Dataset of Diverse Text for Language Modeling. arXiv 2020.](https://arxiv.org/abs/2101.00027)
> >
> > [E] [Maini et al., TOFU: A Task of Fictitious Unlearning for LLMs. arXiv 2024.](https://arxiv.org/abs/2401.06121)
> >
> > [F] [Zhang et al., Negative Preference Optimization: From Catastrophic Collapse to Effective Unlearning. arXiv 2024.](https://arxiv.org/abs/2404.05868)

---

> > > ### Comment · Reviewer_AAXs · 2024-11-25
> > > **Response**
> > >
> > > Thank you for your effort to address my questions. Your detailed revisions and responses have resolved most of my doubts. I have raised my score.

---

### Official Review · Reviewer_sxtU · 2024-10-21

**Soundness:** 3
**Presentation:** 3
**Contribution:** 3
**Rating:** 6
**Confidence:** 3

**Summary:**

The authors identify the limitations of current unlearning methods (e.g., Gradient Ascent (GA)), which can lead to unstable optimization and catastrophic forgetting of retrained knowledge. To overcome these challenges, the paper introduces two novel techniques for robust and efficient unlearning:

1. **Inverted Hinge loss (HIL):** This new loss function suppresses unwanted tokens while maintaining fluency by boosting the probability of the next most likely token.

2. **Fisher-Initialization of Low-rank Adapters (FILA):** Developed through low-rank approximation weighted with relative Fisher information, this method focuses updates on parameters critical for removing targeted knowledge.

The paper demonstrates the effectiveness of these techniques through experiments on the Training Data Extraction Challenge dataset using GPT-Neo models and on the TOFU benchmark with Phi-1.5B and Llama2-7B models. The proposed approach successfully removes sensitive information while maintaining the reasoning and generative capabilities of the models with minimal impact on performance.

In summary, this paper provides innovative solutions to the drawback of GA and demonstrates the effectiveness of the solutions.

**Strengths:**

### Originality
- This paper points out the shortcomings of Gradient Ascent (GA) by analyzing its inverse.
- This paper proposes two strategies to improve these shortcomings.
- This paper demonstrates the effectiveness of their improvements on two datasets.

### Clarity
- The structure of this paper is clear, and most of the content is explained clearly.

### Significance
- This paper provides insights into knowledge unlearning through the analysis of Gradient Ascent (GA).

**Weaknesses:**

- This paper lacks the state-of-the-art knowledge unlearning baselines (such as [1][2]). Although the main goal of the paper is to address the shortcomings of GA, incorporating the state-of-the-art knowledge unlearning for comparison would make it more convincing.

- Some descriptions are not clear enough. For example, lines 221-223 should include more explanation for the reasons. The authors should explain in detail why GA increases the prediction score for all other tokens $v \neq x_t$ in the vocabulary.

- From the experimental results, when only IHL is used, the performance is worse than the original GA. Does this contradict the paper's claim that IHL is designed to address the shortcomings of GA and the analysis of derivatives of GA?

- The paper devotes too much content to the background of knowledge unlearning in the Abstract and in the first paragraph of the Introduction. Since knowledge unlearning is a common problem, I believe it is unnecessary to describe it in such detail. The main content should focus on describing the research conducted in this paper. Specifically, Figure 1 should illustrate the proposed approach rather than the knowledge unlearning problem.

[1] Zhang R, Lin L, Bai Y, et al. Negative preference optimization: From catastrophic collapse to effective unlearning[J]. arXiv preprint arXiv:2404.05868, 2024.

[2] Gao C, Wang L, Weng C, et al. Practical unlearning for large language models[J]. arXiv preprint arXiv:2407.10223, 2024.

**Questions:**

What is the reason for deriving the unlearning mechanism of GA from the formulas in lines 217-220?

---

> ### Author Response · Authors · 2024-11-19
>
> > [W1] This paper lacks the state-of-the-art knowledge unlearning baselines [A, B]. Incorporating the state-of-the-art knowledge unlearning for comparison would make it more convincing.
>
> &rarr; We thank the reviewer for sharing additional baselines.
>
> Regarding the first paper, while Negative Preference Optimization (NPO) alleviates the divergence speed of GA from linear to logarithmic, its backbone unlearning objective still hinges on minimizing the likelihood of $\mathcal{D}_f$ (or maximizing the cross-entropy), and hence NPO still suffers from its divergent behavior. As a result, Figure 5 in [A] shows NPO still leads to suboptimal knowledge retention, as most unlearning trajectories with NPO lead to decreases in model utility of more than 0.1. In comparison, our analogous results in our Figure 5(b) show negligible deterioration in model utility, outperforming NPO on the TOFU benchmark with Llama-2-7b. As [A] has publicly available code, we will revise the draft and add experimental comparisons against NPO.
>
> For the second paper, we are not able to find publicly available code and the paper proposes an unlearning method tailored specifically for the continual unlearning scenario where multiple unlearning requests are made [B]. Because of this difference in experimental setup, we are currently unable to make direct empirical comparisons with our results. For now, we will revise the paper to mention [B] in our related works section.
>
> On a sidenote, please understand that both papers are contemporaneous work, [A] being published in COLM 2024 held this October and [B] an arXiv paper posted this July.
>
> ---
> > [W2] Some descriptions are not clear enough. The authors should explain in detail why GA increases the prediction score for all other tokens in the vocabulary.
>
> &rarr; The increase on all other tokens' scores in GA is due to the cross-entropy loss formulation used in all next-token prediction frameworks. Given a logit value $v_{x}$ for each possible token $x$ in vocabulary set $\mathcal{V}$, the probability of generating the token $x_t$ given the prefix $x_{<t}$ is given by
> $$
> p(x_t | x_{<t}) = \dfrac{\exp(v_{x_t})}{\sum_{x \in \mathcal{V}} \exp(v_x)}
> $$
> Notice there are two ways to decrease $p(x_t | x_{<t})$:
> 1. We can decrease logit value $v_{x_t}$ correponding to the unwanted token $x_t$.
> 2. We can increase the logit value $v_x$ for any possible token $x$ other than $x_t$.
>
> In GA, gradients flow both ways, which leads to equally increasing the logit values of all other possible tokens, and hence the problems discussed in Lines 223-227. We will revise the draft for better clarity.
>
> ---
> > [W3] From the experimental results, when only IHL is used, the performance is worse than the original GA. Does this contradict the paper's claim that IHL is designed to address the shortcomings of GA and the analysis of derivatives of GA?
>
> &rarr; We strongly assert that, the merit of IHL mainly lies in its superior stability in retaining knowledge and generative performance, and that our experimental results are indicative of this strength.
>
> In our TDEC experiments (§4.1), we observe that IHL achieves superior stability (i.e., overcoming catastrophic forgetting during unlearning) compared to GA in most cases. Specifically, when comparing the results in Table 1 between LoRA (using GD) and LoRA+IHL (where GA in GD is replaced by IHL), we find that LoRA+IHL consistently outperforms LoRA in Reasoning, Dialogue, and Pile. Additionally in Figure 3, when comparing the results from GD (blue color) and IHL (orange color), we find that, except for certain cases with GPT-Neo-1.3B (e.g., rank = 32 for Dialogue), IHL outperforms GD in almost all ranks for Reasoning, Dialogue, and the Pile.
>
> Also in our TOFU experiments (§4.3), Figure 5 shows that IHL (green color, replacing GA with IHL) consistently shows negligible decrease in model utility, whereas GD (orange color, using GA) quickly loses its previously acquired knowledge, deviating significantly from the trajectory towards the Retain Set Only oracle (marked as $\star$). Based on these experimental results, we hope the reviewer reconsiders the empirical superiority of IHL over GA.
>
> ---
> > [W4] The paper devotes too much content to the background of knowledge unlearning in the Abstract and in the first paragraph of the Introduction. Figure 1 should illustrate the proposed approach rather than the knowledge unlearning problem.
>
> &rarr; We appreciate the reviewers suggestion. We will revise the abstract and introduction section to reduce the content on general knowledge unlearning, and also replace Figure 1 with a figure that specifically illustrates our proposed methods IHL and FILA.

---

> ### Author Response · Authors · 2024-11-19
>
> > [Q1] What is the reason for deriving the unlearning mechanism of GA from the formulas in lines 217-220?
>
> &rarr; In [C], Gradient Ascent (GA) was demonstrated to successfully unlearn knowledge from large language models (LLMs). Based on these results, GA has become the foundational algorithm in LLM unlearning, often combined with various regularization terms added to maintain general knowledge outside the forget set (See Section 2 of the manuscript). However, the analysis of how GA performs unlearning and the potential issues with this approach have not been adequately addressed. To highlight these concerns, we analyze the derivative of GA, as described in Lines 218-227, to expose the limitations and underlying mechanisms of GA's unlearning process.
>
> In addition to the explanation in Lines 221-227, we clarify how GA performs unlearning during gradient descent: Given the prefix $x_{<t}$, GA reduces the prediction score of the true token $x_t$ by an amount proportional to $1 - p_\theta(x_t | x_{<t})$ and increases the scores of other tokens $v \neq x_t$ by $p_\theta(v | x_{<t})$. This process shifts the predicted token for $x_{<t}$ away from the true token, achieving unlearning.
>
> Based on this analysis, we share three key issues of GA as noted in our manuscript:
> 1. **Gradient Spread:** GA reduces the true token's score while increasing scores of all other tokens. When using large vocabulary sizes as in most LLMs, this leads to gradients primarily boosting tokens other than the true token $x_t$.
> 2. **Unbounded Loss:** With LLMs, GA reduces $\log(p_\theta(x_t | x_{<t}))$ by maximizing the cross-entropy loss, which by nature of entropy, is an unbounded optimization problem. This implies the possibility of divergence as unlearning progresses.
> 3. **Performance Degradation:** Each sequence $x$ in the forget set $\mathcal{D}\_f$ can require different number of model updates for successful unlearning, yet GA applies the same gradient updates to decrease $p_\theta(x_t | x_{<t})$ at every iteration regardless of this distinction. This leads to unnecessary model updates, which we found to induce catastrophic forgetting of knowledge and generative performance.
>
> Our novel objective function IHL is specifically designed to mitigate these limitations (Lines 244-252). We will further clarify this discussion in the revised manuscript.
>
> [A] [Zhang et al., Negative Preference Optimization: From Catastrophic Collapse to Effective Unlearning. arXiv 2024.](https://arxiv.org/abs/2404.05868)
>
> [B] [Gao et al., Practical Unlearning for Large Language Models. arXiv 2024.](https://arxiv.org/abs/2407.10223)
>
> [C] [Jang et al., Knowledge unlearning for mitigating privacy risks in language models. ACL 2023.](https://arxiv.org/abs/2210.01504)

---

> ### Comment · Reviewer_sxtU · 2024-11-24
> **Response**
>
> Thanks for your rebuttal. I decide to maintain my score.

---

> > ### Author Response · Authors · 2024-11-25
> >
> > Thank you for your response. We greatly appreciate the opportunity to address your concerns in our rebuttal.
> >
> > If our rebuttal sufficiently addressed your initial concerns, we kindly ask you to reconsider the scoring in light of the clarifications provided. In case of any additional questions and concerns on our work that led to your decision to maintain your score, we would be grateful if you could share them with us. Your insights are invaluable, and we are committed to improving our work based on your guidance.
> >
> > Sincerely,
> > Authors of Submission11013.

---

> > > ### Comment · Reviewer_sxtU · 2024-11-25
> > > **Response**
> > >
> > > I have read the new version of the paper, and I have no further questions. I will improve my score.

---

### Official Review · Reviewer_T4eT · 2024-10-27

**Soundness:** 3
**Presentation:** 3
**Contribution:** 2
**Rating:** 6
**Confidence:** 3

**Summary:**

The paper works on machine unlearning in LLMs, particularly focusing on the challenges of removing specific data instances from a model's memory without retraining from scratch. The authors propose two strategies: Inverted Hinge Loss (IHL) and Fisher-Initialization of Low-rank Adapters (FILA). IHL is designed to replace the unbounded negative cross-entropy loss in gradient ascent with a more stable and efficient loss function. FILA aims to initialize low-rank adapters in a way that prioritizes the removal of unwanted information and accelerates the unlearning process. Extensive experiments validates that the proposed methods significantly outperform existing baselines in efficiency and post-unlearning performance.

**Strengths:**

1. Authors analyze the derivatives of GA and highlight its shortcomings, the motivation is clear and the theoretical foundation strengthens the rationale for the proposed methods.
2. The introduction of IHL addresses the instability issues of GA by focusing gradient updates on a minimal number of viable replacements for the ground-truth token. This results in a more controlled and stable unlearning process.
3. The proposed strategies are effective. The authors evaluate the methods on multiple datasets and multiple model sizes. This comprehensive evaluation demonstrates the robustness and generalizability of the proposed methods.

**Weaknesses:**

The intuition and connection between the proposed methods, IHL and Fisher-Initialization of FILA, appear somewhat weak. This makes the paper feel like it is stacking two separate tricks rather than offering a unified and coherent approach. A more systematic linkage between these methods would enhance the overall coherence and impact of the paper.

**Questions:**

How robust are the proposed methods to changes in the data distribution of the forget set? For instance, if the forget set contains highly diverse or outlier data, would the unlearning process still be effective?

---

> ### Author Response · Authors · 2024-11-19
>
> > [W1] A more systematic linkage between IHL and FILA would enhance the overall coherence and impact of the paper.
>
> &rarr; We thank the reviewer for bringing this point. We would like to clarify that FILA is specifically designed to cope with the shortcoming of IHL, namely its slow unlearning speed.
>
> As observed in all our experimental results, replacing the negative cross-entropy loss with our IHL leads to superior retention in previously acquired knowledge and generative capabilities, but increases the number of epochs required to fully forget the unlearning targets. Therefore, FILA is designed to accelerate the unlearning process while enjoying the knowledge retention capability of IHL. In Figure 3, note that applying only FILA on top of GD easily leads to significant loss in overall performance, implying that the stability of IHL and the efficiency from FILA form a strong synergy in LoRA-based LLM unlearning.
>
> We will revise the Introduction and Proposed Method sections to clarify this linkage.
>
> ---
> > [Q1] How robust are the proposed methods to changes in the data distribution of the forget set?
>
> &rarr; While limited to forget sets designated by the TDEC and TOFU benchmarks, we believe our experiments demonstrate the robustness of our methods IHL and FILA under varying data diversity. Specifically, the forget sets in TDEC contain sequences from a wide variety of sources such as Github code, the Pile CC, Books3, Freelaw, etc. [A], whereas forget sets in TOFU consist of similarly formatted question-answer pairs on fictitious author profiles. Despite this distinction, our method attains best performance on both setups and hence we can expect our methods to be effective in both scenarios with diversely or uniformly distributed forget set distributions.
>
> [A] [Gao et al., The Pile: An 800GB Dataset of Diverse Text for Language Modeling. arXiv 2020.](https://arxiv.org/abs/2101.00027)

---

> > ### Comment · Reviewer_T4eT · 2024-11-22
> > **Response and a new question**
> >
> > Thank you for your response! I have read your reply and it addresses some of my concerns, which is greatly appreciated! I have a new question: The majority of the experiments in the paper are conducted on models smaller than 3B parameters. Could you provide results for models with 7B parameters or larger, or at least illustrate the trend of performance as the model size scales up?

---

> ### Author Response · Authors · 2024-11-24
> **Response to New Question**
>
> We sincerely thank Reviewer T4eT for the prompt response to our rebuttal. Below is our response to your new question.
>
> ---
> > [Q2] The majority of the experiments in the paper are conducted on models smaller than 3B parameters. Could you provide results for models with 7B parameters or larger, or at least illustrate the trend of performance as the model size scales up?
>
> &rarr; Unfortunately, we are unable to produce results from LLMs larger than 7B at the moment due to limited time and compute resources. Though limited in scale, we can still deduce several insights with respect to the model size, details on which are shared below following our responses on use of larger models.
>
> 1. Scaling TDEC experiments
>     - Please note that for TDEC, we are limited to models publicly known to be pretrained on the Pile dataset [A], as the TDEC dataset contains sequences extracted from the Pile.
>     - Unfortunately, the GPT-Neo model family [B] chosen because of this reason following previous work on GA [C] only scales up to 2.7B. Even though the GPT-NeoX-20B model [D] was developed along the same line of work, this model differs from GPT-Neo in ways other than its model size (e.g., different tokenizer), and thus we are not able to make conclusive observations solely based on the model size.
>     - To the best of our knowledge, OPT [E] is the only open-source family of LLMs pretrained on the Pile with model sizes spanning beyond 7B. However, due to the large time cost required for TDEC evaluation, we are unable to run full set of experiments with either within the rebuttal period. We will add these experiments as future work.
>
> 2. Scaling TOFU experiments
>     - For experiments on the TOFU benchmark [F], recall that we consider the LLM tuned on the TOFU dataset via full-parameter finetuning as our base model. These base models are publicly available only for Phi-1.5B and Llama2-7B via the TOFU repository, results from which we share in Section 4.3.
>     - Therefore, to scale our experiments up to larger models such as Llama2-13B or Llama2-70B, we need to prepare additional base models by full-parameter-finetuning the respective models on TOFU. We found this is not possible under our limited GPU compute.
>     - While we could more efficiently prepare base models via (1) parameter-efficient fine-tuning (e.g. LoRA) or (2) quantization-aware training, both largely deviate from the setup used in the TOFU benchmark, and are likely to introduce confounding factors preventing us to make accurate observations.
>     - Due to this reason, we are unable to extend our TOFU experiments to Llama2-13B and 70B at the moment, but would like to add as future work when given access to larger compute resources.
>
> 3. Insights on model size from current results.
>     - In Table 1, comparing the Reasoning and Dialogue performances across GPT-Neo models of increasing size shows that **preserving language capabilities under LoRA-based unlearning becomes more challenging as the model becomes larger**. For instance, the loss in Reasoning Accuracy of GD worsens from -2.6 to -4.8, then to -6.4 as we increase the model size from 125M, 1.3B, and 2.7B.
>     - We believe this trend is due to **larger models being more likely to memorize pretraining data than smaller models** as demonstrated in previous work [G], and tuning the model to fully forget $\mathcal{D}\_f$ requires more weight perturbations under LoRA tuning, hence the greater loss in previously acquired knowledge. This is also reflected in GPT-Neo-1.3B and 2.7B requiring a larger number of unlearning epochs than GPT-Neo-125M.
>     - Though not exactly comparable, the TOFU results from Phi-1.5B and Llama2-7B (Figure 5) also shows this behavior of larger models. When comparing the unlearning trajectories of GD using the two models, we see that GD increases the forget quality significantly faster in Phi-1.5B than in Llama2-7B.
>     - Despite this difficulty, our IHL+FILA method best minimizes the loss in model utility consistently across all models, which we expect to be the case with LLMs larger than 7B as well.
>
> [A] [Gao et al., The Pile: An 800GB Dataset of Diverse Text for Language Modeling. arXiv 2020.](https://arxiv.org/abs/2101.00027)\
> [B] [Black et al., GPT-Neo: Large Scale Autoregressive Language Modeling with Mesh-Tensorflow.](https://github.com/EleutherAI/gpt-neo)\
> [C] [Jang et al., Knowledge unlearning for mitigating privacy risks in language models. ACL 2023.](https://arxiv.org/abs/2210.01504)\
> [D] [Black et al., GPT-NeoX-20B: An Open-Source Autoregressive Language Model. Workshop at ACL 2022.](https://arxiv.org/abs/2204.06745)\
> [E] [Zhang et al., OPT: Open Pre-trained Transformer Language Models. arXiv 2022.](https://arxiv.org/abs/2205.01068)\
> [F] [Maini et al., TOFU: A Task of Fictitious Unlearning for LLMs. arXiv 2024.](https://arxiv.org/abs/2401.06121)\
> [G] [Carlini et al., Quantifying Memorization Across Neural Language Models. ICLR 2023.](https://arxiv.org/abs/2202.07646)

---

> ### Author Response · Authors · 2024-11-27
> **Gentle Reminder for Reviewer T4eT**
>
> Thank you again for your insightful comments on our work. We are writing to kindly remind the reviewer that we have shared a response to your additional question Q2 above.
>
> As the revision deadline is approaching soon, we would greatly appreciate if you could take the time to review our response, and share any additional feedback or concerns at your earliest convenience. Your insights have been invaluable in refining our work, and we want to ensure we incorporate any suggestions you may have before the deadline. We look forward to your continued guidance.
>
> Sincerely,
>
> Authors of Submission11013.

---

> > ### Comment · Reviewer_T4eT · 2024-11-27
> >
> > Thank you for your efforts to address my concerns. I'm still a little bit unsure about the scaling effect, the current explanation is not strong and with not enough data points. I will maintain my score (and it is already tend to accept), but thanks for your response!

---

### Official Review · Reviewer_EhHm · 2024-11-01

**Soundness:** 2
**Presentation:** 2
**Contribution:** 3
**Rating:** 6
**Confidence:** 3

**Summary:**

The paper proposes a framework to remove sensitive information from LLMs without retraining them from scratch. Recognizing the limitations of common unlearning methods like Gradient Ascent (GA), which risks instability and unintended forgetting, the authors introduce two new techniques. The Inverted Hinge Loss (IHL) method enhances stability by suppressing unwanted tokens with the next most likely alternative, while the Fisher-weighted Initialization of Low-rank Adapters (FILA) uses Fisher information to initialize LoRA adapters, selectively targeting parameters associated with unwanted information to optimize unlearning. This dual approach was evaluated on the Training Data Extraction Challenge and TOFU benchmark with models such as GPT-Neo, Phi-1.5B, and Llama2-7B, achieving efficient unlearning with minimal loss to the model’s reasoning and generative capabilities, and demonstrating improved performance over existing methods.

**Strengths:**

1. The motivation is clearly explained.
2. Extensive experiments have been conducted to prove the effectiveness of the proposed methods.
3. The theoretical analysis strengthens the rationale of the proposed methods.

**Weaknesses:**

1. One of the most significant contributions of this paper is the proposal of Inverse Hard Loss (IHL), which claims to increase the probability of the second-best token only. However, it is not clear why IHL does not affect the probability of other tokens. Based on the definition of IHL in Lines 233, the probability of all other tokens is impacted. As such, IHL can only address problem 1 (Line 224) but cannot address problems 2 and 3 of GA (Lines 224 ~ 226).
2. In Figures 3 and 5, the unlearning performance of employing only IHL (represented in green) does not outperform the GD baseline (depicted in blue), which undermines the effectiveness of IHL.
3. The main results only use GPT-neo models, which are old models. It is better to use more recent models like Llama and Mistral models to make it more practically useful. It is also inconsistent to use different models for main results and analysis.
4. There are no ablations studies for the following settings: 1) full parameter fine-tuning with IHL; 2) LoRA + FILA only; 3) GD + LoRA + FILA.

**Questions:**

1. In Figure 3, why are some data points missing?
2. It is better to add legends in Figures 3 and 4 to improve the clarity.
3. It is better to define “Model Utility” within the paper instead of referring readers to other papers.
4. For the Hinge loss equation in Line 233, since the probability p(.) is in the range of (0,1), the second item within max() function is always larger than 0, right? If so, IHL is to reduce the probability of true tokens but to increase the probability of other tokens, right?

---

> ### Author Response · Authors · 2024-11-19
>
> > [W1] It is not clear why IHL does not affect the probability of other tokens. Based on the definition of IHL (Line 233), the probability of all other tokens is impacted. As such, IHL can only address problem 1 (Line 224) but cannot address problems 2 and 3 of GA (Lines 224-226).
>
> &rarr; To address this question, we will briefly clarify our approach based on the analysis of the derivative of $\mathcal{L}\_{IHL}$ presented in Lines 235–253 of the manuscript. As stated in Lines 249–250, the proposed $\mathcal{L}\_{IHL}$ does not entirely ignore the logits of tokens other than the true token and the second-highest prediction token; instead, it allows these logits to increase at a relatively slower rate.
>
> For example, with the standard $\mathcal{L}\_{GA}$, as shown in the derivative in Line 218, the logits for all tokens except the true token are trained to increase proportionally to their current logit values. In this case, if the logit for the true token $x_t$ decreases, the logits of the other tokens increase even more significantly.
>
> However, the derivative of $\mathcal{L}\_{IHL}$ in Line 238 demonstrates a different learning pattern. Specifically, if unlearning has not yet succeeded (i.e., when $p\_\theta(x_t | x\_{<t})$ is still greater than $p\_\theta(v^* | x\_{<t})$), the derivative for tokens where $v \neq x_t$ and $v \neq v^*$ shows that the gradient for other tokens scales with $p_\theta(v|x_{<t})$ by a factor equal to the difference between $p\_\theta(x_{t}|x_{<t})$ and $p\_\theta(v^{*}| x_{<t})$. This results in only a small fraction of the $p\_\theta(v | x_{<t})$ logit increase. As a result, compared to $\mathcal{L}\_{GA}$, this leads to a relatively slower increase in the logits of other tokens.
>
> ---
> > [W2] In Figures 3 and 5, the unlearning performance of employing only IHL does not outperform the GD baseline, which undermines the effectiveness of IHL.
>
> &rarr; We strongly assert that, the merit of IHL mainly lies in its superior stability in retaining knowledge and generative performance, and that our experimental results are indicative of this strength.
>
> In our TDEC experiments (§4.1), we observe that IHL achieves superior stability (i.e., overcoming catastrophic forgetting during unlearning) compared to GA in most cases. Specifically, when comparing the results in Table 1 between LoRA (using GD) and LoRA+IHL (where GA in GD is replaced by IHL), we find that LoRA+IHL consistently outperforms LoRA in Reasoning, Dialogue, and Pile. Additionally in Figure 3, when comparing the results from GD (blue color) and IHL (orange color), we find that, except for certain cases with GPT-Neo-1.3B (e.g., rank = 32 for Dialogue), IHL outperforms GD in almost all ranks for Reasoning, Dialogue, and the Pile.
>
> Also in our TOFU experiments (§4.3), Figure 5 shows that IHL (green color, replacing GA with IHL) consistently shows negligible decrease in model utility, whereas GD (orange color, using GA) quickly loses its previously acquired knowledge, deviating significantly from the trajectory towards the Retain Set Only oracle (marked as $\star$). Based on these experimental results, we hope the reviewer reconsiders the empirical superiority of IHL over GA.
>
> ---
> > [W3] The main results only use GPT-neo models, which are old models. It is better to use more recent models like Llama and Mistral models to make it more practically useful. It is also inconsistent to use different models for main results and analysis.
>
> &rarr; We would like to clarify that our experimental models are deliberately chosen for meeting unlearning evaluation requirements and maintaining consistency against benchmark standards.
>
> - For our TDEC experiments (§4.1), we choose to use the GPT-Neo family following previous work [A]. Note that because TDEC consists of unlearning targets chosen from the Pile dataset [B], our model choice is limited to those known to be pretrained on the Pile corpus, which excludes more recent larger LLMs such as Llama3.1-8B.
> - For our TOFU experiments (§4.3), we use Phi-1.5B and Llama2-7B models following the original benchmark paper [C]. Not to mention the easy reproducibility due to publicly available base models from which we run unlearning, this also ensures that our empirical findings are directly comparable with original results from the TOFU paper [C] as well as results from papers that experiment unlearning on TOFU [D, E].

---

> ### Author Response · Authors · 2024-11-19
>
> > [W4] There are no ablations studies for the following settings: 1) full parameter fine-tuning with IHL; 2) LoRA + FILA only; 3) GD + LoRA + FILA.
>
> &rarr; We appreciate the reviewer for the suggestions. For further insights, we will add the following ablation studies.
> - Full parameter fine-tuning with IHL as well as other methods.
> - GD + LoRA initialized with FILA. Note this is already presented for TDEC in Figure 3 in green. We will add it to the TOFU experiments as well.
>
> Regarding "2) LoRA + FILA only", could the reviewer clarify which setup this is referring to? It is unclear which objective function the model should be trained on.
>
> ---
> > [Q1] In Figure 3, why are some data points missing?
>
> &rarr; We would like to clarify that no data points are missing in Figure 3: each unlearning method has 5 data points on each plot corresponding to 5 different forget set. It is possible that some cases where unlearning was not successful within 20 epochs (marked with $\times$) appear missing due to large overlaps against successful cases (marked with $\circ$). For better readability, we will adjust Figure 3 to make the data points more distinguishable.
>
> ---
> > [Q2] It is better to add legends in Figures 3 and 4 to improve the clarity.
>
> &rarr; We appreciate the reviewer's suggestion. We will revise Figures 3 and 4 to insert a legend to indicate the method-color correspondence currently covered in the captions.
>
> ---
> > [Q3] It is better to define “Model Utility” within the paper instead of referring readers to other papers.
>
> &rarr; We will add the definition of Model Utility in Section 4.3 for clarity. For reference, Model Utility measures the extent to which the model retains useful information after unlearning. This is done by aggregating (1) the probability of correct answers, (2) ROUGE-L scores, and (3) Truth Ratio of correct answers vs. incorrect answers on questions from three datasets covering the retain set of fictitious authors, real authors, and world facts.
>
> ---
> > [Q4] For the Hinge loss equation in Line 233, since the probability $p(\cdot)$ is in the range of (0,1), the second item within $\max(\cdot)$ function is always larger than 0, right? If so, IHL is to reduce the probability of true tokens but to increase the probability of other tokens, right?
>
> &rarr;  As mentioned in [W1], the reviewer is correct that IHL increases the logits of other tokens when unlearning for $x_t$ is not yet complete (i.e., $p(x\_t | x\_{<t}) > p(v^* | x\_{<t})$). However, it behaves differently once unlearning is achieved (i.e., $p(x_t | x_{<t}) < p(v^* | x_{<t})$). Examining the derivative in Line 238 for tokens $v \neq x_t$ and $v \neq v^*$, we observe that when $p(x_t | x_{<t}) < p(v^* | x_{<t})$, the final gradient becomes positive. As explained in Lines 250–253, this indicates that when unlearning is complete for $x_t$, the logits for other tokens are learned to decrease by a very small fraction of $p(v | x_{<t})$. This unique property not only distinguishes $\mathcal{L}\_{IHL}$ from $\mathcal{L}\_{GA}$ which consistently increases other tokens’ logits, but also reinforces the bounded nature of $\mathcal{L}\_{IHL}$.
>
> We also agree with the reviewer that the second item within $\max(\cdot)$ is always larger than 0, and will thus will revise the formulation of IHL to
> $$
> \mathcal{L}\_{\text{IHL}} = 1 + p\_\theta (x_t | x_{<t}) - \max\_{v \neq x_t}(p\_\theta (v | x_{<t}))
> $$Originally, the outer $\max(\cdot)$ was placed when testing margin terms other than 1, but since using 1 as in the original hinge loss worked best, we adhere to the formulation above.
>
>
> [A] [Jang et al., Knowledge unlearning for mitigating privacy risks in language models. ACL 2023.](https://arxiv.org/abs/2210.01504)
>
> [B] [Gao et al., The Pile: An 800GB Dataset of Diverse Text for Language Modeling. arXiv 2020.](https://arxiv.org/abs/2101.00027)
>
> [C] [Maini et al., TOFU: A Task of Fictitious Unlearning for LLMs. arXiv 2024.](https://arxiv.org/abs/2401.06121)
>
> [D] [Gao et al., Practical Unlearning for Large Language Models. arXiv 2024.](https://arxiv.org/abs/2407.10223)
>
> [E] [Zhang et al., Negative Preference Optimization: From Catastrophic Collapse to Effective Unlearning. arXiv 2024.](https://arxiv.org/abs/2404.05868)

---

> ### Author Response · Authors · 2024-11-26
> **Gentle Reminder for Reviewer EhHm**
>
> Thank you again for your insightful comments on our work. We are writing to kindly remind the reviewer that we have shared a rebuttal and also revised our manuscript to address your concerns and questions.
>
> Please take the time to further review our work, and in case of any remaining concerns or clarifications, we would be grateful if you could share them with us. Your feedback has been invaluable in improving our work, and we look forward to your continued guidance.
>
> Sincerely,
> Authors of Submission11013.

---

> ### Comment · Reviewer_EhHm · 2024-11-27
>
> W1: OK
>
> W2: Apologies for the confusion earlier; the missing legend caused some misunderstanding. Allow me to clarify my question based on the updated version. In Figure 3, it is evident that GD + FILA demonstrates lower accuracy and higher perplexity compared to GD. Similarly, in Figure 5, GD + FILA deviates further from the target (Retain set only) relative to GD. Does this suggest that FILA is ineffective?
>
> W3: I understand that the models were chosen for comparison with benchmarks, but I believe it would be more insightful to demonstrate their effectiveness on newer models.
>
> W4. For “LoRA + FILA only”, I am referring to the loss function GA. In Table 1,  the results of “GD + LoRA initialized with FILA” are also missing. The settings are to demonstrate the effectiveness of using FILA alone.

---

> ### Author Response · Authors · 2024-11-28
> **Follow-up Response to Reviewer EhHm (1/2)**
>
> We sincerely thank Reviewer EhHm for the opportunity to address additional concerns. Our follow-up responses are as below, and we have revised our manuscript to clarify these points as well.
>
> ---
> > [W2] In Figure 3, it is evident that GD + FILA demonstrates lower accuracy and higher perplexity compared to GD. Similarly, in Figure 5, GD + FILA deviates further from the target (Retain set only) relative to GD. Does this suggest that FILA is ineffective?
>
> &rarr; We agree with the reviewer that FILA itself is not effective when combined with GD (which uses GA on $\mathcal{D}_f$ as its unlearning signal). We would like to reiterate that we do not claim FILA to be effective on its own with GD, but our main methodological contribution instead lies in **the synergy between the stability of IHL and efficiency of FILA for LoRA-based unlearning (Lines 258-264)**.
>
> As to why GD + FILA significantly underperforms, this is expected when understanding that FILA's effective role is to initialize adapter weights towards accelerating LoRA-based tuning on $\mathcal{D}_f$ (Lines 264-266). Whether FILA leads to beneficial results in downstream performance depends on which loss function we use. **When using GA loss which can naturally diverge (Lines 220-224), applying FILA makes unlearning diverge even faster, significantly breaking the LLM**. It is only when using with IHL that accelerating unlearning via FILA leads to better results, enjoying the high knowledge retention of IHL alongside the tuning efficiency of FILA. We have revised the manuscript in Lines 419-427 to clarify this point.
>
> ---
> > [W3] I understand that the models were chosen for comparison with benchmarks, but I believe it would be more insightful to demonstrate their effectiveness on newer models.
>
> &rarr; We appreciate the reviewer's suggestion to experiment with newer models, as they can offer additional insights and strenghen our findings from this work.
>
> In our discussion with Reviewer T4eT, we identified that OPT [A], an open-source LLM family (including models larger than 7B trained on the Pile corpus), could serve as another suitable testbed for our TDEC experiments. Similarly, for TOFU experiments, we could explore more recent models such as Mistral [B] and Llama3 [C]. Note that implementing these would require first full-parameter tuning on the TOFU dataset to prepare the base models.
>
> While we would like to conduct and share the results of these additional experiments during the rebuttal period, these setups exceed the time and compute resources currently available to us. Therefore, we plan to explore these directions as future work and kindly ask for the reviewer's understanding of our current constraints.
>
> The experiments included in this work were designed to align with the primary objectives of the study and provide a strong proof of concept, offering a solid foundation for further exploration. We sincerely appreciate the reviewer’s suggestions and will incorporate newer models in future iterations to assess the scalability and broader applicability of our method.
>
> [A] [Zhang et al., OPT: Open Pre-trained Transformer Language Models. arXiv 2022.](https://arxiv.org/abs/2205.01068)\
> [B] [Jiang et al., Mistral 7B. arXiv 2023.](https://arxiv.org/abs/2310.06825)\
> [C] [Meta AI, The Llama 3 Herd of Models. arXiv 2024.](https://arxiv.org/abs/2407.21783)

---

> ### Author Response · Authors · 2024-11-28
> **Follow-up Response to Reviewer EhHm (2/2)**
>
> ---
> > [W4.1]  In Table 1, the results of “GD + LoRA initialized with FILA” are also missing. The settings are to demonstrate the effectiveness of using FILA alone.
>
> &rarr; As mentioned in our response for [W2], the main role of FILA is to accelerate tuning on $\mathcal{D}_f$ via proper initialization, and whether this speed up brings performance benefits after unlearning hinges upon the loss function being optimized for unlearning.
>
> That being said, we have updated our manuscript to include results from GD+FILA in Table 1. Comparing GD vs. GD+FILA, we can see that the number of epochs needed to successfully unlearn $\mathcal{D}_f$ decreases consistently when incorporating FILA with GD, but the downstream performance worsens in many cases (for example, GPT-Neo-1.3B performs worse with GD+FILA on all aspects than just GD). This demonstrates that accelerating the divergent behavior of the GA loss in GD with FILA can worsen downstream performance. On the other hand, FILA becomes beneficial when paired with a stable and bounded objective such as IHL, which is shown in Table 1 with consistent performance boosts from IHL to IHL+FILA.
>
> ---
> > [W4.2] For “LoRA + FILA only”, I am referring to the loss function GA.
>
> &rarr; To clarify, GD is our primary baseline representing GA as (1) GD is equal to GA plus a regularizer on the retain set for better knowledge retention (i.e. $\mathcal{L}\_{\text{GD}} = \mathcal{L}\_{\text{GA}}(\mathcal{D}\_f) + \mathcal{L}\_{\text{LM}}(\mathcal{D}\_r)$) and (2) our preliminary results on TDEC (Table 1) have shown that GD consistently outperforms GA under full-parameter tuning. Notably, GA leads to drastic loss of generative capabilities for GPT-Neo-125M.
>
> For further information, we have run GA+FILA on the TOFU benchmark, results on which can be found in Figure 7 of the Appendix. Similar to GD vs. GD+FILA, we find that GA+FILA suffers from even faster loss in model utility than GA, again reflecting how the divergent behavior is exacerbated with FILA as discussed above.
>
> We are also conducting GA+FILA experiments on our TDEC benchmark. However, due to the high time cost of unlearning evaluation, these results will likely only be available after the rebuttal period. That said, we anticipate that GA+FILA will show similar underperformance as GD+FILA, and will include these findings in the Appendix. We again ask for the reviewer’s understanding of these constraints.

---

> > ### Comment · Reviewer_EhHm · 2024-11-28
> >
> > Thanks for the effort to address my concerns. I would like to raise the score.

---

### Author Response · Authors · 2024-11-19
**Common Response to All Reviewers**

We thank all reviewers for your time and commitment made into reviewing our work. We are deeply encouraged by the overall positive feedback on our valuable motivation towards stable and cost-efficient LLM unlearning [EhHm, T4eT, sxtU, AAXs], strong theoretical foundation [EhHm, T4eT, sxtU], and comprehensive experimentation with promising results [EhHm, T4eT, AAXs].

In light of reviewer-specific questions, our responses and clarifications can be found below in each respective comment. We are in the process of revising our manuscript to reflect the reviewers' comments, and will upload the revised version with a reminder soon. Should any additional questions arise, please share with us and we would be happy to discuss further.

Thank you again for your service.

Sincerely,

Authors of Submission11013.

---

### Author Response · Authors · 2024-11-23
**Revised Manuscript Available for Further Review**

We would like to inform the reviewers that we have updated our manuscript for further review. As our discussion phase is closing within a few days, we respectfully ask the reviewers to review our revised manuscript, and let us know in case of any additional questions or concerns.

For your reference, the revised portions within the manuscript can be found in blue-colored text. Below is our summary of revisions:

1. Writing
    - **Lines 50-75 and 83-87** [sxtU, AAXs]: We have added references to other LLM unlearning methods and brief discussions on them in the Introduction section.
    - **Lines 102-107** [AAXs]: We have added a summary of the main contributions of our paper.
    - **Lines 159-161** [AAXs]: We have added a brief comparison between our study and other relevant studies.
    - **Lines 216-227 and 242-255** [EhHm, sxtU, AAXs]: We have added further details on how the gradients of GA reflects its issues, and how our IHL formulation resolves those issues from a probabilistic perspective.
    - **Lines 258-269** [T4eT]: We have revised the Motivation paragraph of Section 3.4 to clarify the linkage between IHL and FILA.
    - **Lines 270-273** [AAXs]: We have added explanations on the mathematical and intuitive definition of Fisher information.
    - **Lines 420-424** [EhHm, sxtU]: We have clarified the observation that even without FILA, IHL outperforms GD in many cases with both full-parameter and LoRA-based tuning.
    - **Lines 462-465** [EhHm]: We have added details on how the Model Utility metric is measured in TOFU.
    - **Writing flaws** [AAXs]: We thank the reviewer again for spotting these errors. We have corrected the errors in the revision. Additionally, overly long or verbose sentences have been revised for clarity and conciseness, particularly in the Introduction and Related Work sections.
2. Figures and Tables
    - **Figure 1** [sxtU]: We have revised Figure 1 to contain specific illustrations of our IHL and FILA methods in addition to the overall LLM unlearning pipeline.
    - **Table 1** [AAXs]: We have denoted the best performance as bold for each column, and also included color-coded changes in performance after unlearning.
    - **Figures 3 and 4** [EhHm]: We have added legends to both figures and made the $\times$ markers denoting cases where unlearning was unsuccessful larger for better visibility.
3. Additional Experimental Results
    - **IHL with full-finetuning on TDEC (Table 1)** [EhHm]: We have added results from running full-parameter unlearning with IHL on the TDEC dataset. We can see that while GA and GD leads to significant loss in generative performance especially with smaller models (GPT-Neo-125M), IHL exhibits superior stability, minimizing the performance gap with the base model consistently across all GPT-Neo models.
    - **GD+FILA Results on TOFU (Figure 5)** [EhHm]: For completeness, we have added results using FILA with GD to our TOFU experiments, and made the color-coding consistent with Figures 3 and 4 for better readability. Results indicate that the model utility degrades quickly with GD+FILA, as observed in our TDEC experiments.
    - **Comparison vs. KL, DPO, and NPO in TOFU (Figure 7 in Appendix)** [sxtU]: We have run additional experiments on TOFU with three existing baselines: KL [A], DPO [B], and NPO [C]. We found that our IHL+FILA consistently outperforms all three baselines, as all three methods lead to significant degradation in model utility.

[A] [Maini et al., TOFU: A Task of Fictitious Unlearning for LLMs. arXiv 2024.](https://arxiv.org/abs/2401.06121)

[B] [Rafailov et al., Direct Preference Optimization: Your Language Model is Secretly a Reward Model. NeurIPS 2023](https://arxiv.org/abs/2305.18290)

[C] [Zhang et al., Negative Preference Optimization: From Catastrophic Collapse to Effective Unlearning. arXiv 2024.](https://arxiv.org/abs/2404.05868)

---

### Meta-Review · Area_Chair_PgFd · 2024-12-19

**Metareview:**

The paper addresses the challenge of removing sensitive information from Large Language Models (LLMs) without retraining, proposing two techniques: Inverted Hinge Loss (IHL) and Fisher-weighted Initialization of Low-rank Adapters (FILA). IHL aims to stabilize unlearning by focusing on the next most likely token, while FILA uses Fisher information to initialize LoRA adapters, selectively targeting parameters associated with unwanted information to optimize unlearning. The methods were tested on GPT-Neo, Phi-1.5B, and Llama2-7B models, showing efficient unlearning with minimal performance loss. All reviewers agree on the importance of the work and mention that the provided methodology is clearly justified theoretically and empirically. Moreover, the authors have put the effort to modify the manuscript based on the provided reviews and address all their comments. There are suggestions for improvement including clearer explanations (i.e., connection between the two methods are unclear), more recent model evaluations, and better scaling analysis.

**Additional Comments On Reviewer Discussion:**

The original scores were 5 6 5 but after the rebuttal the authors could improve the writing and add extra experimental results to convince the reviewers to increase the scores to 6 6 6.

---

### Decision · Program_Chairs · 2025-01-22

Accept (Poster)